# RADAR: Robust AI-Text Detection via Adversarial Learning

**Xiaomeng Hu**
The Chinese University of Hong Kong
Sha Tin, Hong Kong
xmhu23@cse.cuhk.edu.hk

**Pin-Yu Chen**
IBM Research
New York, USA
pin-yu.chen@ibm.com

**Tsung-Yi Ho**
The Chinese University of Hong Kong
Sha Tin, Hong Kong
tyho@cse.cuhk.edu.hk

## Abstract

Recent advances in large language models (LLMs) and the intensifying popularity of ChatGPT-like applications have blurred the boundary of high-quality text generation between humans and machines. However, in addition to the anticipated revolutionary changes to our technology and society, the difficulty of distinguishing LLM-generated texts (AI-text) from human-generated texts poses new challenges of misuse and fairness, such as fake content generation, plagiarism, and false accusations of innocent writers. While existing works show that current AI-text detectors are not robust to LLM-based paraphrasing, this paper aims to bridge this gap by proposing a new framework called RADAR, which jointly trains a robust AI-text detector via adversarial learning. RADAR is based on adversarial training of a paraphraser and a detector. The paraphraser's goal is to generate realistic content to evade AI-text detection. RADAR uses the feedback from the detector to update the paraphraser, and vice versa. Evaluated with 8 different LLMs (Pythia, Dolly 2.0, Palmyra, Camel, GPT-J, Dolly 1.0, LLaMA, and Vicuna) across 4 datasets, experimental results show that RADAR significantly outperforms existing AI-text detection methods, especially when paraphrasing is in place. We also identify the strong transferability of RADAR from instruction-tuned LLMs to other LLMs, and evaluate the improved capability of RADAR via GPT-3.5-Turbo.

> **Project Page and Demos:** https://radar.vizhub.ai
> IBM demo is developed by Hendrik Strobelt and Benjamin Hoover at IBM Research
> HuggingFace demo is developed by Xiaomeng Hu

## 1 Introduction

Large language models (LLMs) are high-capacity neural networks that are pretrained at web-scale datasets. They are foundation models achieving state-of-the-art performance in a wide range of natural language processing tasks (e.g. document completion, question answering, machine translation, and content creation with text prompts) with advanced capabilities such as in-context learning and reasoning (e.g. chain of thoughts). In particular, LLMs are the backbone of many ChatGPT-like conversational bots that enable text generation with high fluency and accuracy. However, while LLMs and their derived applications are expected to become ubiquitous in our future technology and society,

37th Conference on Neural Information Processing Systems (NeurIPS 2023).

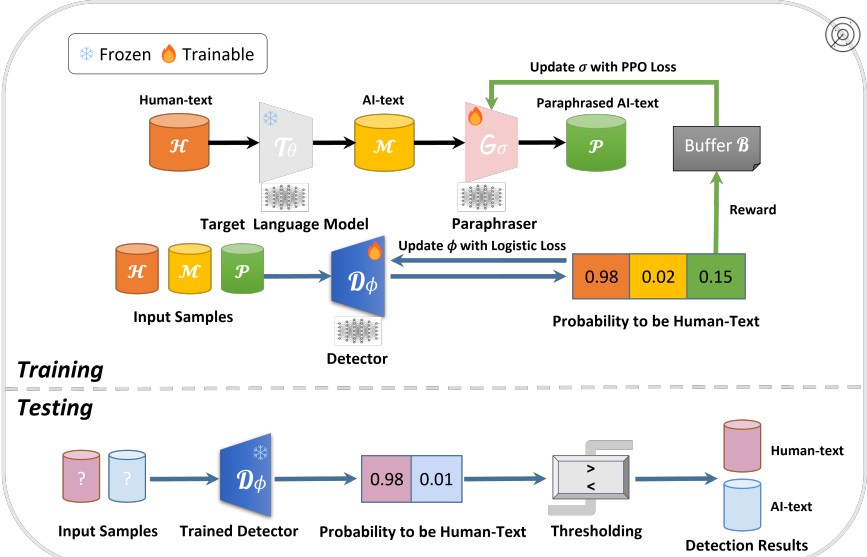

Figure 1: Overview of RADAR. An AI-text corpus is first generated from a target (frozen) language model from a human-text corpus. In RADAR, we introduce a paraphraser (a tunable language model) and a detector (a separate tunable language model). In the training stage, the detector aims to discern human-text v.s. AI-text, while the paraphraser aims to rewrite AI-text to evade detection. The model parameters of the paraphraser and the detector are updated in an adversarial learning manner as described in Section 3. In the evaluation stage, the trained detector is deployed to predict the likelihood of AI-generated content for any input instance.

new risks in failing to distinguish the so-called "AI text" generated by LLMs have emerged and gained considerable attention due to various reasons. The problem of reliable AI-text detection is motivated by realistic socio-technological challenges such as fake content generation, AI plagiarism (e.g. using LLMs for writing tests), and false accusations of innocent writers. According to a report released by OpenAI[1], their latest AI-text detector is admittedly not fully reliable. In the reported evaluation of some challenging cases for English texts, their classifier only correctly identifies 26% of AI-text (true positives) while incorrectly classifying 9% of human-written text (false positives). Moreover, a recent study [21] found that state-of-the-art AI-text detectors demonstrated severely degraded performance when encountering texts written by non-native English speakers.

What can be even more challenging in AI-text detection is that existing AI-text detectors are prone to be manipulated. The authors in [27, 17] showed that using LLMs as a paraphraser can easily evade several AI-text detection methods, even in the scenario when the original AI-text had been watermarked. These findings sparked a heated debate about whether and how we can successfully design a reliable AI-text detector. While [27] theoretically quantifies the best detector's performance with respect to the total variation distance between AI-text and human-text distributions and argues that AI-text is difficult to detect, another work [3] proves that it is possible to obtain a reliable AI-text detector unless the human-text distribution is exactly the same as the AI-text distribution, based on an information-theoretical analysis (i.e., the sample complexity of Chernoff information and likelihood-ratio-based detectors).

To improve AI-text detection, we propose **RADAR**, a framework for training a robust AI-text detector using adversarial learning. An overview of RADAR is illustrated in Figure 1. Our proposal draws inspiration from adversarial machine learning techniques that train a high-quality generator by introducing a discriminator to form a two-player game, such as generative adversarial networks (GANs) [10]. In RADAR, we introduce a paraphraser and a detector as two players with opposite objectives. The paraphraser's goal is to generate realistic content that can evade AI-text detection, while the detector's goal is to enhance AI-text detectability. In our framework, both the paraphraser and the detector are parametrized by separate LLMs. During training, the paraphraser learns to rewrite the text from a training corpus (generated by a target LLM from a human-text corpus) with the

[1] https://openai.com/blog/new-ai-classifier-for-indicating-ai-written-text

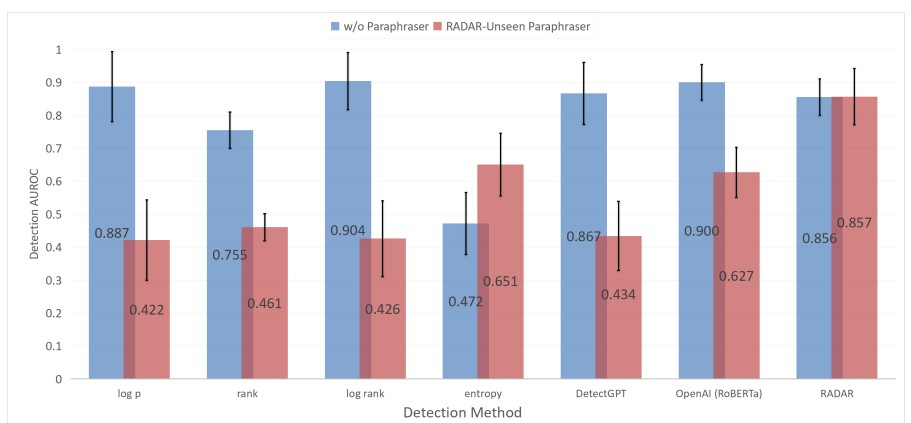

Figure 2: Performance evaluation (AUROC) of 8 LLMs over 4 human-text datasets. *w/o paraphraser* means the evaluation with the original AI-text corpora (the yellow bin $\mathcal{M}$ in Figure 1). *RADAR-Unseen paraphraser* means the evaluation with the paraphrased AI-text (the green bin $\mathcal{P}$ in Figure 1) generated from an independent paraphraser (OpenAI's GPT-3.5-Turbo API) that is not used in RADAR. The black error bar represents the standard deviation of the detection AUROCs across 8 LLMs. Please refer to Section 4.2 for more implementation details.

aim of decreasing the likelihood of AI-text prediction by the detector, whereas the detector aims to enhance the detection performance by learning to compare human-text v.s. AI-text from the training data and the paraphraser's output. These two players iteratively update their model parameters until their respective validation loss becomes stable. Specifically, the paraphraser treats the prediction of the detector as a reward and uses Proximal Policy Optimization (PPO) [28] for updates. The detector updates its parameters based on a logistic loss function evaluated on the human-text and AI-text corpora (including the texts generated by the paraphraser). In the evaluation phase, the trained detector is deployed to predict the likelihood of AI-written content for any input instance. When compared with 6 existing detectors, our experimental results on 8 different LLMs and 4 datasets show that RADAR attains similar detection performance on the original AI-generated texts (a relatively easy task) and simultaneously improves the AI-text detectability when facing an "unseen" paraphraser (i.e. this paraphraser is not used in RADAR). The result is summarized in Figure 2. When facing an unseen paraphraser (GPT-3.5-Turbo), the area under the receiver operating characteristic (AUROC) score of RADAR is improved by 31.64% compared to the best existing detector, suggesting a significant improvement and reliable AI-text detection power enabled by RADAR.

We summarize our **main contributions** as follows:

- To the best of our knowledge, RADAR is the first study that leverages the idea of adversarial learning between a paraphraser and a detector for training a robust AI-text detector.

- The experiments on 8 different LLMs (Pythia, Dolly 2.0, Palmyra, Camel, GPT-J, Dolly 1.0, LLaMA, and Vicuna) and 4 datasets show that unlike the six existing supervised and unsupervised AI-text detection methods, RADAR is the only robust detector that attains consistently high detection performance. RADAR's detector is not weakened by paraphrasing, as shown in Figure 2.

- We also find the strong transferability of RADAR's detection capability. The detectors of RADAR obtained from instruction-tuned first-class LLMs (e.g., Vicuna-7B) are also effective on other LLMs, suggesting the possibility of training a universal AI-text detector based on the state-of-the-art LLMs.

## 2 Related Work

**AI-Text Detection.** The research in AI-text detection can be divided into three approaches. (i) Statistical methods: some statistics such as entropy [19], n-gram frequency, and perplexity are used as a threshold to discern AI-text. A typical example is GLTR [8], which exploits entropy, probability, and probability rank for detection. A more recent work is DetectGPT [23], which assumes that the machine-generated text always lies in the negative curvature region of the log probability of the LLM of interest. Based on this hypothesis, DetectGPT perturbs the input text with a mask-filling language

model, such as T5 [25]. Then, AI-text detection is performed by comparing the log probability of the text and its infilled variants. (ii) Classification methods: AI-text detection is formulated as a binary classification task, and a classifier is trained for a target language model [37, 29, 26, 13]. For example, OpenAI trains its AI-text classifier with a RoBERTa-based model [29].

The developers collected samples from the WebText dataset[2] and labeled them as human-generated. Then, for each target GPT-2 model, they collected the generated samples and labeled them as machine-generated. Finally, they fine-tuned the pretrained RoBERTa-based model [29] for AI-text classification. More recently, with the appearance of CharGPT, OpenAI tuned a GPT model called AI-Classifier[1] using data from several sources. The human-written text comes from three sources: a new Wikipedia dataset, the WebText dataset collected in 2019, and a set of human demonstrations collected as part of training InstructGPT [24]. To collect machine-generated text, for the Wikipedia and WebText datasets, they truncated the articles sampled from the original corpus and used 34 models to generate article completion, pairing each generated text with the original article. For the demonstrations, they used a model to generate responses for each prompt and paired them with the corresponding human demonstrations. This detector was only accessible via a web interface since its release in January 2023, and it has been taken down since July 2023. (iii) Watermark methods: post-hoc watermarking techniques, such as rule-based methods [1, 15, 31] and deep-learning-based methods [6, 32], can be applied to an LLM. At inference time, [16] proposed a soft watermarking scheme to embed a watermark in each word of the generated sentence by dividing the vocabulary into different lists and sampling the next token in a differentiated manner. However, many existing AI-text detectors are shown to be significantly weakened by paraphrasing in [27].

**Adversarial Learning for Natural Language Generation.** The success of GAN [10] in the computer vision domain has motivated many studies in natural language generation. However, since text generation is a sequential sampling process that occurs in a discrete vocabulary space, it is difficult to directly train a text generator using back-propagation in an end-to-end manner [36, 7, 5, 35]. There are two common approaches to tackle this problem. The first one is to replace the discrete sampling operation with continuous approximation techniques [35, 5], such as Gumbel-Softmax [14, 22]. The second one is to view text generation as a decision-making process and cast the generator as a policy [36, 34, 7, 33]. A typical example is SeqGAN [36]. During generation, SeqGAN considers the generated tokens as the state and the next token to be generated as the action, and it adopts Monte Carlo search to collect reward signals from the discriminator. Instead of using a classifier as the discriminator, the Diversity-Promoting GAN [34] uses a unidirectional LSTM as the discriminator and combines both word-level and sentence-level rewards into training. TextGAIL [33] proposed an imitation learning paradigm in which the rewards of the human-written text are regarded as a constant value. Then, both the rewards from human-text and AI-text are used to optimize the generator with PPO. These works all used warm-up training for the generator with maximum likelihood estimation (MLE) on the probability of the generated text sequence. On the other hand, [7] trained a language GAN from scratch. Our proposed RADAR differs from these works in that we focus on training a robust AI-text detector with a tunable paraphraser. Another line of work, such as [20, 4], uses paraphrasing techniques to find adversarial examples for natural language processing tasks and for training a robust language model via adversarial training. Their focus is on the correctness of natural language understanding, which is beyond our scope of AI-text detection.

## 3 RADAR: Methodology and Algorithms

We start this section by giving an overview and mathematical notations of our proposed RADAR framework in Figure 1. Then, in Sections 3.1 and 3.2, we provide the details on the design and training of the paraphraser and detector, respectively. Finally, we will summarise the entire training process into an algorithmic procedure in Section 3.3.

**High-Level Methodology.** Our RADAR framework consists of three neural-network-based language models (LMs): the target LM $\mathcal{T}_\theta$, the detector $\mathcal{D}_\phi$ and the paraphraser $\mathcal{G}_\sigma$, parameterized with $\theta$, $\phi$ and $\sigma$, respectively. We note that $\mathcal{T}_\theta$ is frozen (no updates on $\theta$) in the entire process. We summarize RADAR into three key steps:

- **Step 1 (Data preparation):** Before training, we build $\mathcal{M}$, the corpus of AI-text, by applying document completion based on the prefix span of text in the human-text corpus $\mathcal{H}$ using $\mathcal{T}_\theta$.

---

[2] `https://huggingface.co/datasets/openwebtext`

- **Step 2 (Paraphraser update):** We collect AI-text samples $x_m$ from $\mathcal{M}$ and use $\mathcal{G}_\phi$ to do paraphrasing on $x_m$ to generate paraphrased AI-text $x_p$ to form a corpus $\mathcal{P}$. Then, we use the reward of $x_p$ returned by the detector $\mathcal{D}_\theta$ to update the paraphraser $\mathcal{G}_\phi$ using PPO.

- **Step 3 (Dectector update):** We use the human-text samples $x_h$ from $\mathcal{H}$, the original AI-text samples $x_m$ from $\mathcal{M}$, and the paraphrased AI-text samples $x_p$ from $\mathcal{P}$ in step 2 to update the detector $\mathcal{D}_\theta$ with a logistic loss function.

- **Step 4 (Performance Validation and Evaluation):** During training, we use the test set of WebText as the validation dataset to estimate RADAR's performance. For evaluation, we use $\mathcal{T}_\theta$ to generate AI-text for the evaluation dataset and to calculate RADAR's detection AUROC.

Step 2 to Step 3 can be repeated until there is no improvement in the AUROC evaluated on the validation dataset. The nature of rivalry in adversarial learning and the introduced competition helps the detector to learn to be robust in detecting both original and paraphrased AI-text.

## 3.1 Training Paraphraser via Clipped PPO with Entropy Penalty

In RADAR, the goal of the paraphraser $\mathcal{G}_\sigma$ is to paraphrase the input machine-generated text $x_m$. We model the generation of paraphrased text as a decision-making process, taking $x_m$ as the state and the output text $x_p$ as the action. In particular, we optimize $\mathcal{G}_\sigma$ using the reward feedback from the detector $\mathcal{D}_\phi$ with PPO. The output of $\mathcal{D}_\phi(x_p)$ is the predicted likelihood of $x_p$ being Human-text. The reward returned by $x_p$ and the log probability of the text $x_p$ are defined in Eq. 1:

$$R(x_p, \phi) = \mathcal{D}_\phi(x_p) \in [0, 1]; \quad \log P_{\mathcal{G}_\sigma}(x_p|x_m) = \sum_{i=1}^{N} \log P_{\mathcal{G}_\sigma}(x_p^i|x_m, x_p^{1:i-1}), \tag{1}$$

where $x_p^i$ means the $i$-th token in the sentence $x_p$ of length $N$ and $x_p^{1:i-1}$ represents the first $i-1$ tokens in $x_p$ ($x_p^{1:0}$ means the default starting token).

We propose Clipped PPO with Entropy Penalty (`cppo-ep`) in RADAR to optimize $\mathcal{G}_\sigma$. Let $\text{clip}(\cdot, a, b)$ denote a value-clipping operation with a lower limit $a$ and an upper limit $b$, $r(\sigma, x_m, x_p)$ be the importance sampling ratio between a new policy $\mathcal{G}_\sigma$ and an old policy $\mathcal{G}_{\sigma'}$, and $(x_m, x_p) \sim P_{\mathcal{G}_{\sigma'}}$ be a state-action pair sampled from $\mathcal{G}_{\sigma'}$. The loss of `cppo-ep` is defined as:

$$L_{\mathcal{G}}(\sigma) = \mathbb{E}_{(x_m, x_p) \sim P_{\mathcal{G}_{\sigma'}}} \underbrace{-\min\{\text{clip}(r(\sigma, x_m, x_p), 1-\epsilon, 1+\epsilon), r(\sigma, x_m, x_p)\} \cdot A(x_p, \phi)}_{L_\mathcal{A}} \underbrace{-\gamma S(\sigma)}_{L_\mathcal{E}} \tag{2}$$

where $\mathbb{E}$ denotes expectation, $\epsilon$ is a parameter used in clipping to avoid the importance ratio $r$ from being too large, $A(x_p, \phi)$ is the advantage item of the paraphrased text $x_p$ obtained by applying normalization to $R(x_p, \phi)$ across the entire PPO sample buffer $\mathcal{B}$. $S(\sigma) = \mathbb{E}_{(x_m, x_p) \sim P_{\mathcal{G}_{\sigma'}}} - P_{\mathcal{G}_\sigma}(x_p|x_m) \log P_{\mathcal{G}_\sigma}(x_p|x_m)$, which is an entropy term introduced to encourage $\mathcal{G}_\sigma$ to explore more diverse generation policy. $\gamma$ is a coefficient to control the ratio between $L_\mathcal{A}$ and $L_\mathcal{E}$, in order to make a balance between advantage ($L_\mathcal{A}$) and diversity ($L_\mathcal{E}$) when paraphrasing.

## 3.2 Training Detector via Reweighted Logistic Loss

In a typical GAN training process, the discriminator receives an equal amount of positive and negative samples in each step, assuring an in-batch sample balance. However, in RADAR, by construction, the number of AI-text samples is twice the number of human-text samples, because each $x_h$ from the human-text corpus $\mathcal{H}$ is paired with a sample $x_m$ from the original AI-text corpus $\mathcal{M}$ as well as a paraphrased sample $x_p$ generated by the paraphraser $\mathcal{G}_\phi$. To handle this in-batch imbalance problem, we use a reweighted logistic loss function to optimize the detector $D_\phi$, as described in Eq. 3:

$$L_\mathcal{D}(\phi) = \underbrace{-\mathbb{E}_{x_h \sim \mathcal{H}} \log \mathcal{D}_\phi(x_h)}_{L_\mathcal{H}: \text{loss on human-text}} + \lambda \underbrace{\mathbb{E}_{x_m \sim \mathcal{M}} - \log(1 - \mathcal{D}_\phi(x_m))}_{L_\mathcal{M}^1: \text{loss on original AI-text}} + \lambda \underbrace{\mathbb{E}_{x_m \sim \mathcal{M}} - \log(1 - \mathcal{D}_\phi(\mathcal{G}_\sigma(x_m)))}_{L_\mathcal{M}^2: \text{loss on paraphrased AI-text}}$$

$$\tag{3}$$

Recall that $\mathcal{D}_\phi(x) \in [0, 1]$ is the predicted probability of an input instance $x$ being Human-text. $L_\mathcal{H}$ is the loss to improve the correctness of predicting $x_h \sim \mathcal{H}$ as human-written. $L_\mathcal{M} = L_\mathcal{M}^1 + L_\mathcal{M}^2$, where $L_\mathcal{M}^1$ and $L_\mathcal{M}^2$ are used to avoid $x_m$ and $x_p$ from being predicted as human-text, respectively. $\lambda$ is a coefficient ranging from 0 to 1. We introduce $\lambda$ to adjust the proportion of AI-text components in the overall loss function to alleviate the effects of sample imbalance.

### 3.3 RADAR Algorithm

The entire training procedure of RADAR is summarized in Algorithm 1. For a given target LLM, RADAR returns a trained paraphraser and a trained detector through the designed training steps. In the evaluation phase, the detector is used to predict the likelihood of AI-text for any input instance.

---

**Algorithm 1** RADAR: Robust AI-Text Detection via Adversarial Learning

---

1: **Data initialization:**
2: Collect human-written text to build human-text corpus $\mathcal{H}$
3: Select a target language model $\mathcal{T}_\theta$ to perform document completion on $\mathcal{H}$ to build the corresponding AI-text corpus $\mathcal{M}$
4: Build a replay buffer $\mathcal{B}$ to store samples temporarily collected for training
5: Build a validation dataset $\mathcal{V}$ from $\mathcal{H}$ and $\mathcal{M}$
6: **Model initialization:**
7: Detector $\mathcal{D}_\phi \leftarrow \phi_{\text{pretrain}}$ (a pretrained language model)
8: Paraphraser $\mathcal{G}_\sigma \leftarrow \sigma_{\text{pretrain}}$ (a pretrained language model)
9: **Model training:**
10: **for** $i = 1$ : maximum step **do**
11:     Sample $x_h$ and its corresponding $x_m$ from $\mathcal{H}$ and $\mathcal{M}$ respectively
12:     Use $\mathcal{G}_\sigma$ to paraphrase $x_m$ and generate $x_p$
13:     Collect reward $R(x_p, \phi)$ as in Eq. 1
14:     Normalize $R(x_p, \phi)$ to compute the advantage function $A(x_p, \phi)$ used in Eq. 2
15:     Fill $\mathcal{B}$ with $(x_h, x_m, x_p, A(x_p, \phi))$
16:     $\sigma' \leftarrow \sigma$  # initialize the old policy $\sigma'$ as the current policy $\sigma$
17:     **for** $(x_h, x_m, x_p, A(x_p, \phi)) \in \mathcal{B}$ **do**
18:         Compute the log probability $\log P_{\mathcal{G}_\sigma}(x_p|x_m)$ and $\log P_{\mathcal{G}_{\sigma'}}(x_p|x_m)$ using Eq. 1
19:         Update $\mathcal{G}_\sigma$ using Eq. 2
20:     **end for**
21:     **for** $(x_h, x_m, x_p, A(x_p, \phi)) \in \mathcal{B}$ **do**
22:         Update $\mathcal{D}_\phi$ using Eq. 3
23:     **end for**
24:     Clear $\mathcal{B}$
25:     Evaluate AUROC of $\mathcal{D}_\phi$ on the validation dataset $\mathcal{V}$
26: **end for**
27: Detector $\mathcal{D}_\phi \leftarrow \phi_{\text{best}}$ (the detector model with the best AUROC on the validation dataset)
28: Paraphraser $\mathcal{G}_\sigma \leftarrow \sigma_{\text{best}}$ (the paraphraser model which pairs with $\phi_{\text{best}}$)
29: Return $\mathcal{D}_\phi$ and $\mathcal{G}_\sigma$

---

## 4 Experiments

### 4.1 Experimen Setup

**Datasets and Metrics.** For training, we sampled 160K documents from WebText [9] to build the human-text corpus $\mathcal{H}$. Then, we build the original AI-text corpus $\mathcal{M}$ from $\mathcal{H}$ using a target language model $\mathcal{T}_\theta$, which performs text completion using the first 30 tokens as the prompt and limits the sentence length to be 200 tokens. For evaluation, we select four human-text datasets covering different domains. Following [23], we use Xsum, SQuAD, and Reddit WritingPrompts (WP) to test a detector's ability to detect fake news, avoid academic fraud, and identify machine-generated literature innovation, respectively. In addition, we also use the non-native-authored TOEFL dataset (TOFEL) [21] to evaluate a detector's bias when encountering non-native-authored English text. Please see Appendix A for more details about the evaluation datasets. Following existing works, we report the area under the receiver operating characteristic curve (AUROC) score by varying the detector's threshold as the performance measure (higher is better), which captures the relationship between the true positive rate and the false positive rate.

**Comparisons.** We compare RADAR with various detection methods. These methods include the OpenAI (RoBERTa) model which is fine-tuned on WebText [9] and GPT-2 [2] generations, as well as the statistical approaches including log probability, rank, log rank, entropy, and DetectGPT [8, 19, 23].

Table 1: Summary of the studied large language models

| Parameter Count | Model Name | Organization | Pretrain Data | Instruction Fine-tune Data |
|---|---|---|---|---|
| 3B | Pythia-2.8B | EleutherAI | The Pile [3] | ✗ |
| | Dolly-V2-3B | Databricks | | databricks-dolly-15k[4] |
| 5B | Palmyra-base | Writer | Writer's custom dataset | ✗ |
| | Camel-5B | Writer | | 70K instruction-response records by Writer Linguist team |
| 6B | GPT-J-6B | EleutherAI | The Pile | ✗ |
| | Dolly-V1-6B | Databricks | | Standford Alpaca 52K instruction-following demonstrations[5] |
| 7B | LLaMA-7B | Meta | Various sources[6] | ✗ |
| | Vicuna-7B | LMsys | | 70K conversations collected from ShareGPT [7] |

Specifically, we implemented DetectGPT using the trained T5-large model as the mask-filling model and performed 10 perturbations for each sentence to be detected.

**Large Language Models.** For the target LLM $\mathcal{T}_\theta$, we select 4 pairs of LLMs and summarize them in Table 1. Each pair contains an open-source LLM and its fine-tuned version via instruction-tuning.

**Paraphrase Configurations.** We consider two settings: *without (w/o) paraphrasing* and *with paraphrasing*. To prepare the machine-generated text for evaluation, for the w/o paraphrasing setting, we use the original AI-text corpus $\mathcal{M}$ generated by a target LLM based on an evaluation dataset. For the with paraphrasing setting, we define two types of paraphrasing: *seen paraphraser* and *unseen paraphraser*. The seen paraphraser refers to the paraphraser $\mathcal{G}_\sigma$ returned by RADAR. The unseen paraphraser means a new paraphraser that has not participated in training the detector of RADAR. We used the OpenAI API service of GPT-3.5-Turbo as the default unseen paraphraser. The prompt we used for paraphrasing is "Enhance word choices to make the sentence sound more like a human", as inspired by [21].

**Implementation Details.** We provide the detailed setups when implementing Algorithm 1. We build a PPO buffer $\mathcal{B}$ that can temporarily store 256 pairs of data for subsequent training. We use the pre-trained T5-large and RoBERTa-large models as the initialization of $\mathcal{G}_\sigma$ and $\mathcal{D}_\phi$ respectively. During training, we set the batch size to 32 and train the models until the validation loss converges. We use AdamW as the optimizer with the initial learning rate set to 1e-5 and use linear decay for both $\mathcal{G}_\sigma$ and $\mathcal{D}_\phi$. We set $\lambda = 0.5$ for sample balancing in Eq. 3 and set $\gamma = 0.01$ in Eq. 2. We follow the same construction principle of the training dataset to create the 4 evaluation datasets based on Xsum, SQuAD, WP, and TOFEL. Experiments were run on 2 GPUS (NVIDIA Tesla V100 32GB).

## 4.2 Performance Evaluation and Comparison with Existing Methods

We run three groups of experiments (w/o paraphraser, seen paraphraser, and unseen paraphraser) and report the overall results of RADAR and the compared methods on all 4 datasets in Table 2. The reported AUROC scores are averaged over the 8 considered LLMs. In the relatively easy case of without paraphrasing, most detectors attain good AUROC scores. RADAR attains a comparable performance (0.856) to the best existing detector (log rank, 0.904). The slightly worse performance of RADAR can be explained by the tradeoff in enhancing AI-text detection against paraphrasing.

When facing paraphrasing, all existing methods except entropy show significant performance degradation. The drop in AUROC compared to the w/o paraphrasing case ranges from $10.4\%$ to $81.7\%$. While entropy is shown to be more robust to paraphrasing, its AUROC score can be quite low. On the contrary, RADAR demonstrates robust and superior detection power, attaining the best performance on every dataset. As shown in Figure 2, the average AUROC score of RADAR (0.857) improves the best existing method (entropy, 0.651) by 31.64% against the unseen paraphraser. On average, RADAR is more robust to the seen paraphraser than the unseen paraphraser, because the seen paraphraser is what is used to train the detector in RADAR. More importantly, the detection performance of RADAR is stable across different paraphrasing schema, suggesting that RADAR can successfully mitigate the performance drop in AI-text detection.

---

[3]https://huggingface.co/datasets/EleutherAI/pile

[4]https://huggingface.co/datasets/databricks/databricks-dolly-15k

[5]https://github.com/tatsu-lab/stanford_alpaca/blame/main/alpaca_data.json

[6]Collected from CCNet [67%], C4 [15%], GitHub [4.5%], Wikipedia [4.5%], Books [4.5%], ArXiv [2.5%], Stack Exchange [2%]

[7]https://sharegpt.com/

Table 2: AUROC score averaged over 8 target LLMs. RADAR-Seen Paraphraser means the paraphraser used in RADAR ($\mathcal{G}_\sigma$). RADAR-Unseen Paraphraser is OpenAI's GPT-3.5-Turbo API. The notations {①,②} denote the best/second-best method for each dataset.

| Evaluation Schema | Method | Evaluation Dataset | | | | |
|---|---|---|---|---|---|---|
| | | Xsum | SQuAD | WP | TOFEL | Average |
| w/o Paraphraser | log p | 0.882 | 0.868 | 0.967② | 0.832 | 0.887 |
| | rank | 0.722 | 0.752 | 0.814 | 0.731 | 0.755 |
| | log rank | 0.902 | 0.893② | 0.975① | 0.847② | 0.904① |
| | entropy | 0.536 | 0.521 | 0.296 | 0.534 | 0.472 |
| | DetectGPT | 0.874 | 0.790 | 0.883 | 0.919① | 0.867 |
| | OpenAI (RoBERTa) | 0.953① | 0.914① | 0.924 | 0.810 | 0.900② |
| | RADAR | 0.934② | 0.825 | 0.847 | 0.820 | 0.856 |
| RADAR-Seen Paraphraser | log p | 0.230 | 0.156 | 0.275 | 0.130 | 0.198 |
| | rank | 0.334 | 0.282 | 0.357 | 0.163 | 0.284 |
| | log rank | 0.245 | 0.175 | 0.281 | 0.134 | 0.209 |
| | entropy | 0.796 | 0.845② | 0.763 | 0.876② | 0.820② |
| | DetectGPT | 0.191 | 0.105 | 0.117 | 0.177 | 0.159 |
| | OpenAI (RoBERTa) | 0.821② | 0.842 | 0.892② | 0.670 | 0.806 |
| | RADAR | 0.920① | 0.927① | 0.908① | 0.932① | 0.922① |
| RADAR-Unseen Paraphraser | log p | 0.266 | 0.343 | 0.641 | 0.438 | 0.422 |
| | rank | 0.433 | 0.436 | 0.632 | 0.342 | 0.461 |
| | log rank | 0.282 | 0.371 | 0.632 | 0.421 | 0.426 |
| | entropy | 0.779 | 0.710② | 0.499 | 0.618 | 0.651② |
| | DetectGPT | 0.360 | 0.384 | 0.609 | 0.630② | 0.434 |
| | OpenAI (RoBERTa) | 0.789② | 0.629 | 0.726② | 0.364 | 0.627 |
| | RADAR | 0.955① | 0.861① | 0.851① | 0.763① | 0.857① |

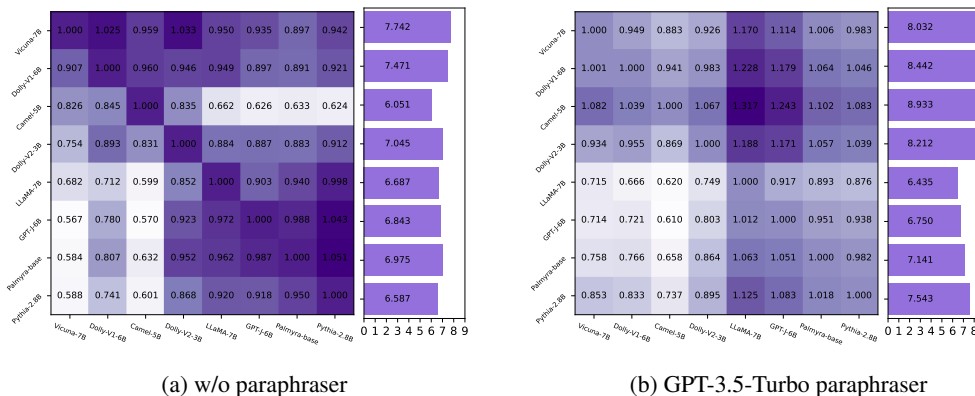

(a) w/o paraphraser                (b) GPT-3.5-Turbo paraphraser

Figure 3: RADAR's detection transferability between pairs of 8 LLMs in Table 1. In the matrix, each row is the source LLM (model A) for training the detector, and each column is the target LLM (model B) for evaluation. The reported value in the matrix represents the detection transferability from A to B. A larger value indicates better transferability. The bar chart shows the row-wise sum of the matrix, indicating the holistic transferability of each source LLM.

### 4.3 AI-Text Detection Transferability of RADAR

We explore the AI-text detection transferability of RADAR between the 8 LLMs and report the ratio F(A,B)=AUROC(A,B)/AUROC(B,B) for each LLM pair (A,B), where AUROC(A,B) means using the RADAR's detector trained on model A to evaluate the AI-text generated by model B. A larger ratio means better transferability from A to B. Figure 3 shows the matrix of pairwise detection transferability and the bar chart of the holistic detection transferability to all the 8 LLMs in the without and unseen paraphrasing settings. We highlight two key observations as follows.

**(I) Instruction-tuned models have better detection transferability.** Partitioning the LLMs into two groups, we can find that the detector targeting an instruction-tuned LLM (top 4 rows) generally transfers better than the detector targeting the corresponding LLM without instruction-tuning (bottom 4 rows). Take the pair (Vicuna-7B, LLaMA-7B) as an example, we can see that without paraphrasing, F(Vicuna-7B,LLaMA) can reach up to 95.0%. On the other hand, F(LLaMA-7B,Vicuna-7B) can only

account for $68.2\%$. Sorting the detectors according to the holistic detection transferalbility (which is presented in the bar chart), we can see the top-3 detectors are all trained with the instruction-tuned LLMs. A similar conclusion can be made for the with paraphrasing setting. Moreover, there is no obvious trend between the target LLM size and the resulting detection performance. The effect of instruction tuning on transferability is more prominent than model size.

**(II) RADAR achieves better detection transferability against paraphrasing.** Another interesting finding is that RADAR's transferability is generally improved when paraphrasing is in place. Comparing the two bar charts in Fig. 3a and Fig. 3b, the average holistic detection transferability (over all LLMs) is increased by $11.6\%$. Except for LLaMA-7B (3.8% drop) and GPT-J-6B (1.4% drop), all other LLMs' holistic transferability scores are improved from 2.4% (Palmyra-base) to 47.6% (Camel-5B).

**Transfer detection on AI-text generated by GPT-4.** We also test RADAR detectors on the texts generated by GPT-4. The results show that 5 out of 8 RADAR models can outperform the OpenAI (RoBERTa), and three of them can achieve more than 0.8 detection AUROC. For example, RADAR trained on Camel-5B can achieve 0.915 detection AUROC on GPT-4 generations. The results show that the RADAR can achieve good transfer detection for GPT-4. The details are given in Appendix K.

**Ensemble detection.** We also explored whether and how ensemble learning benefits detection by combining the outputs of detectors. The results show that the detection performance can be lifted by carefully tuning the ensemble ratio and the model to be combined. Please see Appendix G for the exact experiment results.

To sum up, we believe our findings suggest promising results for training a universal robust AI-text detector by leveraging state-of-the-art LLMs, and RADAR can use a smaller-sized and weaker LLM to achieve good detection performance on texts generated from top-notching LLMs (such as GPT-4).

## 4.4 Variants of Paraphrasing

In addition to paraphrasing the original LLM-generated texts, we also evaluate the detection performance when paraphrasing human texts (the output is labeled as AI-text). We also allow paraphrasing multiple times in our analysis. We conduct our experiments on the Xsum dataset using the detector trained with Camel-5B. The paraphraser for evaluation is GPT-3.5-Turbo. As shown in Figure 4a, we find that RADAR is the only detector robust to multi-round paraphrasing. On paraphrased AI-text, all existing methods suffer from a notable performance drop. On paraphrased human-text, RADAR remains effective, along with two existing methods (OpenAI (RoBERTa) and entropy). In general, multi-round paraphrasing does not seem to increase the difficulty of AI-text detection. We also find RADAR is robust to Dipper [18], another paraphrase model. Please see Appendix I for details.

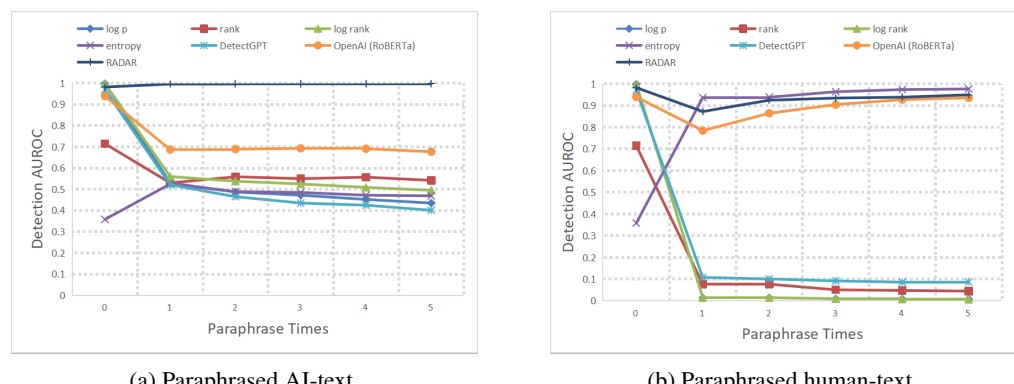

|                      |                      |
|:--------------------:|:--------------------:|
| (a) Paraphrased AI-text | (b) Paraphrased human-text |

Figure 4: Detection AUROC of RADAR against multiple paraphrasing. The experiments are conducted on Xsum using the detector trained for Camel-5B.

## 4.5 Evaluation on RADAR's Paraphraser

Although our focus is on training a robust AI-text detector via RADAR, as a by-product, we expect to obtain a better paraphraser through adversarial learning. To verify this hypothesis, we compare the quality of the initial paraphraser (a pretrained LLM) and the final paraphraser returned by RADAR

using GPT-3.5-Turbo's response. We select 100 documents from WebText [9] and use 4 different paraphrasers from RADAR to paraphrase the documents. Then, we ask GPT-3.5-Turbo to rate sentences generated by these paraphrasers versus their initial version (T5-large). Figure 5a shows that RADAR also improves the quality of paraphrasing. Figure 5b shows that the RADAR's paraphraser can score higher if it is trained with a larger target LLM with instruction tuning. Following [11, 12], we also evaluate RADAR's paraphrasers on Quora Question Pairs (QQP[8]) and use iBLEU ($\alpha = 0.8$) [30] as the metric (higher is better). Figure 5c shows that the paraphrasing performance can be improved via RADAR as all the RADAR-paraphrasers can achieve a larger iBLEU score than T5-large.

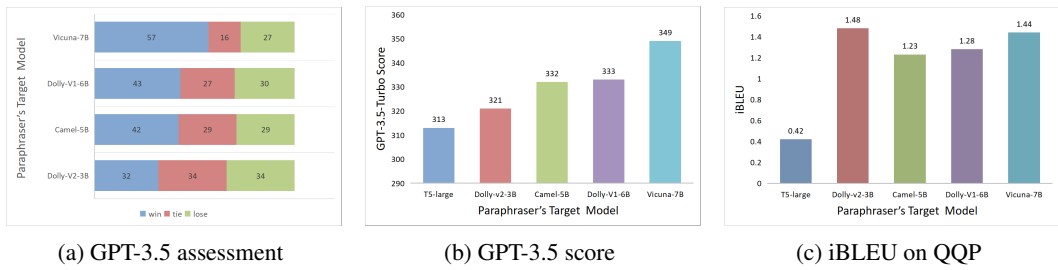

(a) GPT-3.5 assessment          (b) GPT-3.5 score          (c) iBLEU on QQP

Figure 5: Evaluation of RADAR's paraphraser versus its initial version (T5-large).

### 4.6   Balancing the Detection Performance in the with and without Paraphrasing Settings

From Figure 2, we can observe that though RADAR can achieve robust detection under paraphrasing, it is (slightly) worse than some of the existing baselines when AI-text data are unperturbed (i.e., w/o paraphrasing). We run a trade-off analysis on the weight coefficient $\lambda$ in Equation (3) to study whether RADAR can be further tuned to achieve competitive performance on unperturbed data while still being robust to paraphrasing. We use Vicuna-7B as the target model to train 10 RADAR detectors by varying $\lambda$ from 0.1 to 1.0 with 0.1 increment, and then evaluate these detectors as well as other detection baselines on the evaluation datasets. The results in Appendix J show that we can promote RADAR's performance on unperturbed data while still preserving high detection AUROC on paraphrased data. Take $\lambda = 0.6$ as an example. When we change $\lambda$ from 0.5 (the default value of $\lambda$) to 0.6, the AUROC of w/o paraphrasing increases from 0.906 to 0.937, while the AUROC of unseen-paraphrasing also increases from 0.892 to 0.920. The result suggests that the detection performance of RADAR in the with and without paraphrasing settings can be simultaneously improved or better balanced with careful tuning of the hyperparameter $\lambda$ during training.

## 5   Conclusion

In this paper, we presented a robust AI-text detector training framework called RADAR, which adopts adversarial learning to jointly train a detector and a paraphraser. RADAR addresses the shortcoming of existing detectors when facing LLM-paraphrased texts. Our extensive experiments on 8 LLMs and 4 datasets validated the effectiveness of RADAR and demonstrated its strong transferability across LLMs. We believe our results shed new light on improving AI-text detection.

## 6   Limitations and Ethical Considerations

While RADAR is more robust to paraphrasing than existing baselines measured on 4 datasets, sometimes it may show degraded detection performance against native LLM-generated texts (without paraphrasing) when compared to the best existing detection method. Moreover, like every existing AI-text detector, we acknowledge that our detector is not perfect and will likely give incorrect predictions in some cases. In terms of ethical considerations, we suggest users use our tool to assist with identifying AI-written content at scale and with discretion. If the detection result is to be used as evidence, further validation steps are necessary as RADAR cannot always make correct predictions.

---

[8]https://www.kaggle.com/c/quora-question-pairs/data

## Acknowledgement

The authors thank James Sanders and Jonathon Hartley for providing examples that can weaken the detection of RADAR models.

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

# Appendix

## A  Human-text Corpora

We summarize the human-text corpora we used in RADAR's training, validation, and evaluation phases in Table A1. It shows the usage of these corpora, the source where they come from, and the number of samples we select from them for evaluation.

Table A1: Summary of the used human-text corpora

| Phase | Source Dataset | Dataset Key | Sample Counts |
|---|---|---|---|
| Trainng | WebText-train | text | 160000 |
| Validation | WebText-test | text | 4007 |
| Evaluation | Xsum-train | document | 500 |
| | SQuAD-train | context | 312 |
| | WritingPrompts-train | document | 500 |
| | TOFEL-test | text | 90 |

## B  Details of Existing Detectors

Every detector assigns a score to the given text and determines whether the text is generated by AI based on the score. We introduce the scores used in existing detectors in the following.

**Unsupervised Methods.** In this paper, we leverage log-p, rank, log-rank, and entropy as the baselines. They are all unsupervised methods. They depend on statistical metrics of the given text to determine if it is an AI-text. Specifically, they input the given text to the target language model $\mathcal{T}_\theta$, and sniff statistics from $\mathcal{T}_\theta$'s output. We use $M_{\text{log-p}}$, $M_{\text{rank}}$, $M_{\text{log-rank}}$, and $M_{\text{entropy}}$ to represent the score for them respectively. These scores are calculated as below:

$$M_{\text{log-p}}(x) = \frac{1}{N-1} \sum_{i=2}^{N} \log P_{\mathcal{T}_\theta}(x^i | x^{1:i-1}) \tag{A1}$$

$$M_{\text{rank}}(x) = -\frac{1}{N-1} \sum_{i=2}^{N} \text{index}(\text{sort}(\log P_{\mathcal{T}_\theta}(\cdot | x^{1:i-1})), x^i) \tag{A2}$$

$$M_{\text{log-rank}}(x) = -\frac{1}{N-1} \sum_{i=2}^{N} \log(\text{index}(\text{sort}(\log P_{\mathcal{T}_\theta}(\cdot | x^{1:i-1})), x^i)) \tag{A3}$$

$$M_{\text{entropy}}(x) = -\frac{1}{N-1} \sum_{i=2}^{N} \sum_{j=1}^{C} P_{\mathcal{T}_\theta}(j | x^{1:i-1}) \log P_{\mathcal{T}_\theta}(j | x^{1:i-1}) \tag{A4}$$

where $N$ is the length of the input sentence $x$, $C$ is the size of the vocabulary, $x_i$ means the $i$-th token in $x$ and $x^{1:i-1}$ represents the first $i-1$ tokens in $x$, sort($a$) is a sorting operation which inputs a list $a$ and returns a new list in descending order, index($a$,$b$) is a indexing operation which inputs a list $a$ with an element $b$ and outputs the position of $b$ in $a$.

DetectGPT is also an unsupervised detection method we compare with in this paper. It introduces another language model (denoted as $\mathcal{G}_\sigma$), which is used to do perturbations on the given text. DetectGPT uses the perturbation discrepancy as the score assigned to the text (denoted as $M_{\text{DetectGPT}}$), which is shown below:

$$M_{\text{DetectGPT}}(x) = \frac{\log P_{\mathcal{T}_\theta}(x) - \tilde{u}}{\tilde{\sigma}_x} \tag{A5}$$

where $\tilde{u} = \frac{1}{k} \sum_{i=1}^{k} \log P_{\mathcal{T}_\theta}(\tilde{x}_i)$, $\tilde{\sigma}_x^2 = \frac{1}{k-1} \sum_{i=1}^{k} (\log P_{\mathcal{T}_\theta}(\tilde{x}_i) - \tilde{u})^2$, $\tilde{x}_i \sim P_{\mathcal{G}_\sigma}(\cdot | x_i)$, and $k$ is the number of perturbations applied on each text.

**Supervised Methods.** RADAR, as well as OpenAI (RoBERTa), are both supervised detection methods. Let $D$ denote the OpenAI (RoBERTa) or RADAR's detector. The score $M_D$ that they assigned to the input text $x$ is defined below:

$$M_D(x) = \text{Softmax}(f_D(x))[0] \tag{A6}$$

where $f_D(x)$ means $D$'s [CLS] token's logits over the whole label set. $\text{Softmax}(f_D(x))[0]$ means the prediction probability of AI-text, and $\text{Softmax}(f_D(x))[1]$ means the prediction probability of human-text. Since there are only two labels (AI-text vs human-text) in the label set, the detection is equivalent to a logistic regression task with a scalar output.

## C  RADAR Loss Visualization

We visualize RADAR's training process by presenting the training loss and validation performance below. We take the Camel-5B language model as an example.

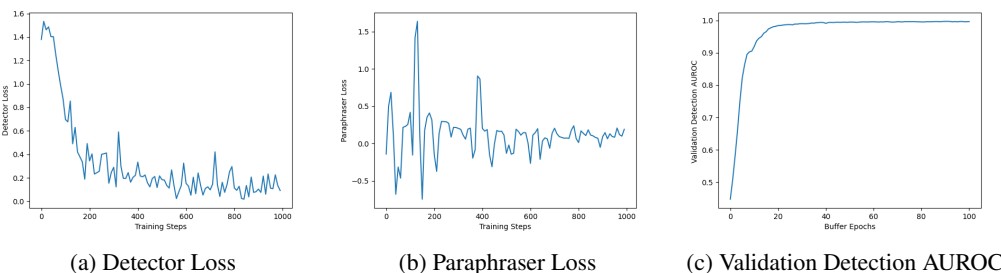

(a) Detector Loss                    (b) Paraphraser Loss          (c) Validation Detection AUROC

Figure A1: Visulization of the training process of RADAR targeting Camel-5B.

From Figure A1a and Figure A1b we can see that the loss for both the Detector and the Paraphraser converge. From Figure A1c we can conclude that RADAR's detection capacity on the validation dataset is promoted to a stable state as the training loss of the detector and the paraphraser converge.

## D  Paraphrase Settings

When using a RADAR-seen paraphraser, the input is: *Paraphrase: [s]*, Where [s] is the slot for the input AI-text. The paraphraser adopts top-k sampling and nucleus sampling strategy to decode a new word. Top-k sampling only focuses on the $k$ highest-probability tokens. Nucleus sampling sorts the sampling distribution in descending order and selects top-n tokens from the sorted distribution until their cumulative probability exceeds $p$, and then samples the next token from the top-n candidates. In our experiment, we set $k$ to 50 and $p$ to 0.95.

When using GPT-3.5-Turbo's API service as another paraphrasing tool (RADAR-Unseen paraphraser) to paraphrase the AI-texts, the instruction we input is *Enhance the word choices in the sentence to sound more like that of a human*. When paraphrasing the human-texts, the instruction is *Worsen the word choices in the sentence to sound less like that of a human*.

Multi-round paraphrasing can be easily achieved by using the paraphrased text as input text and using the paraphraser to re-paraphrase it.

## E  Complete Experimental Results

We show all the evaluation results in Table A2 (without paraphraser), Table A3 (with RADAR-Unseen paraphraser) and Table A4 (with RADAR-Seen paraphraser).

## F  Case Study for RADAR

**Sample Selection**. For each evaluation dataset (Xsum, SQuAD, WP, and TOFEL), we randomly select one sample from the human-text corpora and use 8 instruction-tuned LLMs (Vicuna-7B,

Table A2: Performance evaluation (AUROC) with no paraphraser.

| Method | Dataset | Target Language Model | | | | | | | |
|---|---|---|---|---|---|---|---|---|---|
| | | Pythia-2.8B | Dolly-V2-3B | Palmyra-base | Camel-5B | GPT-J-6B | Dolly-V1-6B | LLaMA-7B | Vicuna-7B |
| log p | Xsum | 0.829 | 0.993 | 0.903 | 0.998 | 0.809 | 0.937 | 0.612 | 0.977 |
| | SQuAD | 0.805 | 0.992 | 0.907 | 0.999 | 0.779 | 0.916 | 0.610 | 0.932 |
| | WP | 0.948 | 0.997 | 0.967 | 0.999 | 0.942 | 0.985 | 0.900 | 0.996 |
| | TOFEL | 0.705 | 0.965 | 0.770 | 0.999 | 0.718 | 0.886 | 0.644 | 0.971 |
| rank | Xsum | 0.758 | 0.794 | 0.734 | 0.715 | 0.746 | 0.756 | 0.610 | 0.659 |
| | SQuAD | 0.789 | 0.831 | 0.796 | 0.752 | 0.787 | 0.790 | 0.627 | 0.645 |
| | WP | 0.821 | 0.859 | 0.818 | 0.825 | 0.826 | 0.841 | 0.758 | 0.761 |
| | TOFEL | 0.739 | 0.792 | 0.768 | 0.766 | 0.726 | 0.754 | 0.657 | 0.647 |
| log rank | Xsum | 0.875 | 0.991 | 0.928 | 0.998 | 0.849 | 0.944 | 0.667 | 0.962 |
| | SQuAD | 0.864 | 0.991 | 0.942 | 0.999 | 0.833 | 0.932 | 0.669 | 0.917 |
| | WP | 0.963 | 0.996 | 0.977 | 0.999 | 0.957 | 0.987 | 0.924 | 0.994 |
| | TOFEL | 0.740 | 0.960 | 0.807 | 0.995 | 0.749 | 0.889 | 0.675 | 0.959 |
| entropy | Xsum | 0.613 | 0.514 | 0.529 | 0.358 | 0.600 | 0.522 | 0.704 | 0.445 |
| | SQuAD | 0.614 | 0.506 | 0.515 | 0.358 | 0.573 | 0.527 | 0.651 | 0.422 |
| | WP | 0.364 | 0.284 | 0.308 | 0.183 | 0.344 | 0.294 | 0.421 | 0.173 |
| | TOFEL | 0.617 | 0.505 | 0.628 | 0.455 | 0.616 | 0.535 | 0.607 | 0.311 |
| DetectGPT | Xsum | 0.871 | 0.989 | 0.909 | 0.970 | 0.806 | 0.837 | 0.709 | 0.900 |
| | SQuAD | 0.774 | 0.976 | 0.854 | 0.966 | 0.702 | 0.697 | 0.561 | 0.788 |
| | WP | 0.859 | 0.988 | 0.906 | 0.967 | 0.813 | 0.849 | 0.726 | 0.959 |
| | TOFEL | 0.863 | 0.999 | 0.927 | 0.989 | 0.875 | 0.899 | 0.824 | 0.976 |
| OpenAI (RoBERTa) | Xsum | 0.991 | 0.962 | 0.993 | 0.939 | 0.979 | 0.980 | 0.900 | 0.880 |
| | SQuAD | 0.974 | 0.898 | 0.978 | 0.880 | 0.925 | 0.948 | 0.839 | 0.871 |
| | WP | 0.983 | 0.887 | 0.988 | 0.869 | 0.958 | 0.951 | 0.900 | 0.856 |
| | TOFEL | 0.887 | 0.747 | 0.915 | 0.748 | 0.872 | 0.851 | 0.710 | 0.748 |
| RADAR | Xsum | 0.955 | 0.873 | 0.979 | 0.982 | 0.923 | 0.926 | 0.916 | 0.916 |
| | SQuAD | 0.821 | 0.762 | 0.849 | 0.968 | 0.743 | 0.829 | 0.725 | 0.902 |
| | WP | 0.845 | 0.697 | 0.896 | 0.965 | 0.763 | 0.899 | 0.810 | 0.899 |
| | TOFEL | 0.796 | 0.742 | 0.938 | 0.763 | 0.852 | 0.781 | 0.779 | 0.908 |

Table A3: Performance evaluation (AUROC) with RADAR-Unseen Paraphraser.

| Method | Dataset | Target Language Model | | | | | | | |
|---|---|---|---|---|---|---|---|---|---|
| | | Pythia-2.8B | Dolly-V2-3B | Palmyra-base | Camel-5B | GPT-J-6B | Dolly-V1-6B | LLaMA-7B | Vicuna-7B |
| log p | Xsum | 0.173 | 0.285 | 0.212 | 0.530 | 0.141 | 0.237 | 0.144 | 0.408 |
| | SQuAD | 0.247 | 0.413 | 0.328 | 0.550 | 0.241 | 0.305 | 0.238 | 0.425 |
| | WP | 0.572 | 0.665 | 0.603 | 0.772 | 0.562 | 0.641 | 0.546 | 0.767 |
| | TOFEL | 0.256 | 0.436 | 0.305 | 0.701 | 0.295 | 0.442 | 0.413 | 0.653 |
| rank | Xsum | 0.429 | 0.429 | 0.478 | 0.530 | 0.382 | 0.437 | 0.414 | 0.368 |
| | SQuAD | 0.42 | 0.463 | 0.480 | 0.486 | 0.400 | 0.431 | 0.410 | 0.395 |
| | WP | 0.685 | 0.676 | 0.691 | 0.662 | 0.580 | 0.693 | 0.497 | 0.569 |
| | TOFEL | 0.323 | 0.377 | 0.310 | 0.397 | 0.335 | 0.366 | 0.300 | 0.326 |
| log rank | Xsum | 0.201 | 0.293 | 0.248 | 0.560 | 0.164 | 0.264 | 0.150 | 0.376 |
| | SQuAD | 0.289 | 0.426 | 0.372 | 0.595 | 0.272 | 0.340 | 0.245 | 0.425 |
| | WP | 0.579 | 0.644 | 0.611 | 0.771 | 0.569 | 0.629 | 0.525 | 0.731 |
| | TOFEL | 0.263 | 0.405 | 0.305 | 0.673 | 0.296 | 0.445 | 0.376 | 0.602 |
| entropy | Xsum | 0.846 | 0.743 | 0.804 | 0.525 | 0.881 | 0.788 | 0.898 | 0.747 |
| | SQuAD | 0.801 | 0.650 | 0.726 | 0.537 | 0.778 | 0.725 | 0.808 | 0.651 |
| | WP | 0.556 | 0.513 | 0.545 | 0.306 | 0.573 | 0.512 | 0.587 | 0.396 |
| | TOFEL | 0.735 | 0.597 | 0.737 | 0.530 | 0.739 | 0.577 | 0.590 | 0.436 |
| DetectGPT | Xsum | 0.260 | 0.389 | 0.291 | 0.521 | 0.217 | 0.254 | 0.213 | 0.374 |
| | SQuAD | 0.288 | 0.447 | 0.343 | 0.554 | 0.252 | 0.288 | 0.204 | 0.311 |
| | WP | 0.404 | 0.518 | 0.439 | 0.652 | 0.393 | 0.393 | 0.305 | 0.550 |
| | TOFEL | 0.459 | 0.662 | 0.573 | 0.827 | 0.572 | 0.618 | 0.582 | 0.746 |
| OpenAI (RoBERTa) | Xsum | 0.885 | 0.809 | 0.899 | 0.687 | 0.824 | 0.764 | 0.774 | 0.668 |
| | SQuAD | 0.727 | 0.597 | 0.757 | 0.486 | 0.665 | 0.638 | 0.612 | 0.552 |
| | WP | 0.791 | 0.708 | 0.806 | 0.596 | 0.780 | 0.748 | 0.718 | 0.663 |
| | TOFEL | 0.453 | 0.265 | 0.511 | 0.308 | 0.388 | 0.309 | 0.348 | 0.332 |
| RADAR | Xsum | 0.966 | 0.957 | 0.976 | 0.996 | 0.904 | 0.985 | 0.910 | 0.947 |
| | SQuAD | 0.872 | 0.873 | 0.873 | 0.988 | 0.763 | 0.934 | 0.698 | 0.887 |
| | WP | 0.843 | 0.898 | 0.824 | 0.989 | 0.726 | 0.965 | 0.655 | 0.907 |
| | TOFEL | 0.784 | 0.775 | 0.772 | 0.918 | 0.634 | 0.809 | 0.584 | 0.827 |

Dolly-V1-6B, Camel-5B, Dolly-V2-3B, LLaMA-7B, GPT-J-6B, Palmyra-base, Pythia-2.8B) to generate completions for each text, then use GPT-3.5-Turbo API service to perform paraphrase to all completions. Thus, we get 64 AI-texts in total (32 completions, 32 paraphrases).

**Case Selection**. Each AI-text has a source model, for the detection of one given text, we use the RADAR-detector trained for its source model. Then, we collect the one with the largest confidence to be machine-generated (most likely to be correctly detected) and the one with the smallest confidence to be machine-generated (mostly likely to evade detection) for the following case study.

**Analysis**. We show two detection cases in Table A5. One with a higher probability (0.9999) is an easy-to-detect case, and another with a lower probability (0.0031) is a difficult-to-detect case. Specifically, the latter can only be detected when our detection threshold drops below 0.0031. From the table, we can see that the misclassification has a good explanation, because the AI-text is nearly identical to the original Human-text, except for the inclusion of several additional words. In fact, the AI-text can be seen as the Human-text with a suffix composed of several words appended to it.

Table A4: Performance evaluation (AUROC) with RADAR-Seen Paraphraser. Red text means the evaluating method is not capable to perform detection for some samples in the evaluating dataset and the reported value is the detection AUROC after removing these samples .

| Method | Dataset | Target Language Model | | | | | | | |
|---|---|---|---|---|---|---|---|---|---|
| | | Pythia-2.8B | Dolly-V2-3B | Palmyra-base | Camel-5B | GPT-J-6B | Dolly-V1-6B | LLaMA-7B | Vicuna-7B |
| log p | Xsum | 0.269 | 0.191 | 0.287 | 0.328 | 0.217 | 0.174 | 0.142 | 0.228 |
| | SQuAD | 0.279 | 0.057 | 0.248 | 0.233 | 0.158 | 0.065 | 0.081 | 0.130 |
| | WP | 0.555 | 0.165 | 0.347 | 0.296 | 0.271 | 0.182 | 0.179 | 0.203 |
| | TOEFL | 0.251 | 0.025 | 0.204 | 0.153 | 0.172 | 0.052 | 0.087 | 0.099 |
| rank | Xsum | 0.429 | 0.261 | 0.410 | 0.387 | 0.340 | 0.251 | 0.324 | 0.267 |
| | SQuAD | 0.502 | 0.149 | 0.377 | 0.301 | 0.293 | 0.132 | 0.259 | 0.243 |
| | WP | 0.631 | 0.258 | 0.439 | 0.329 | 0.333 | 0.267 | 0.291 | 0.305 |
| | TOEFL | 0.326 | 0.042 | 0.215 | 0.140 | 0.239 | 0.062 | 0.177 | 0.106 |
| log rank | Xsum | 0.306 | 0.197 | 0.320 | 0.339 | 0.241 | 0.189 | 0.154 | 0.214 |
| | SQuAD | 0.328 | 0.064 | 0.280 | 0.244 | 0.184 | 0.075 | 0.091 | 0.134 |
| | WP | 0.582 | 0.167 | 0.366 | 0.292 | 0.282 | 0.182 | 0.187 | 0.187 |
| | TOEFL | 0.267 | 0.025 | 0.227 | 0.144 | 0.181 | 0.057 | 0.085 | 0.085 |
| entropy | Xsum | 0.778 | 0.783 | 0.772 | 0.647 | 0.814 | 0.851 | 0.871 | 0.848 |
| | SQuAD | 0.771 | 0.903 | 0.769 | 0.728 | 0.850 | 0.930 | 0.913 | 0.895 |
| | WP | 0.612 | 0.835 | 0.708 | 0.693 | 0.760 | 0.840 | 0.804 | 0.852 |
| | TOEFL | 0.813 | 0.952 | 0.825 | 0.793 | 0.879 | 0.935 | 0.891 | 0.922 |
| DetectGPT | Xsum | 0.270 | 0.214 | 0.280 | 0.284 | 0.224 | 0.129 | 0.078 | 0.143 |
| | SQuAD | 0.231 | 0.093 | 0.201 | 0.181 | 0.111 | 0.030 | 0.027 | 0.043 |
| | WP | 0.258 | 0.135 | 0.192 | 0.223 | 0.073 | 0.061 | 0.025 | 0.066 |
| | TOEFL | 0.327 | 0.107 | 0.322 | 0.237 | 0.187 | 0.081 | 0.040 | 0.111 |
| OpenAI (RoBERTa) | Xsum | 0.844 | 0.839 | 0.829 | 0.716 | 0.786 | 0.830 | 0.883 | 0.842 |
| | SQuAD | 0.851 | 0.905 | 0.871 | 0.758 | 0.830 | 0.835 | 0.847 | 0.838 |
| | WP | 0.887 | 0.903 | 0.918 | 0.856 | 0.901 | 0.898 | 0.895 | 0.881 |
| | TOEFL | 0.711 | 0.621 | 0.692 | 0.610 | 0.716 | 0.681 | 0.681 | 0.649 |
| RADAR | Xsum | 0.879 | 0.920 | 0.913 | 0.913 | 0.890 | 0.945 | 0.972 | 0.924 |
| | SQuAD | 0.917 | 0.926 | 0.917 | 0.856 | 0.904 | 0.953 | 0.988 | 0.958 |
| | WP | 0.873 | 0.868 | 0.953 | 0.724 | 0.974 | 0.959 | 0.990 | 0.925 |
| | TOEFL | 0.969 | 0.894 | 0.926 | 0.809 | 0.952 | 0.947 | 0.994 | 0.964 |

Table A5: Case study of RADAR's detection performance.

| Source Model | Source Dataset | Original Human Text | Paraphraser | Text | Probability to be AI-text |
|---|---|---|---|---|---|
| Camel-5B | WP | Aliens visit earth and are fascinated by other animals but find humans completely unremarkable. Hello, let me begin this report by stating just how thankful I am to have been sent to Earth. It was truly an eye opening experience and without your guidance and funding w--ould have been utterly impossible. Let me give you an overview of how Earth functions, and I think you will be surprised to find that it works very similar to our pl--anet. There exists a large number of organisms going through one stage or another of change while they fall into their respective places on the planet. The conclus--ion of my research has confirmed what we have beli--eved to be a universal law called net impact theory, w--hich can be described in two parts: 1. a species can o--nly exist if it has a net positive impact on its environ--ment. 2. Species' which detract from their environme--nt will | GPT-3.5-Turbo | Greetings I would like to express my utmost gratitude for the extraordinary experience I had while visiting our neighboring planet Mars. It has been a decade sin--ce I last scrutinized Mars' features and the informati--on gathered during my journey has revitalized my fa--scination for this captivating planet. Despite the tech--nological advancements in Mars exploration I was st--ill able to witness the landscape in its natural state ca--pturing breathtaking photographs and filming videos that showcased the red planet's impressive terrain fasc--inating geology and intriguing scientific discoveries. This experience has forever changed my perception of humanity and inspired my passion for space exploratio--n hoping to further contribute to our understanding of the universe and our place within it. | 0.9999 |
| Camel-5B | Xsum | Maj Richard Scott, 40, is accused of driving at speeds of up to 95mph (153km/h) in bad weather before the smash on a B-road in Wiltshire. Gareth Hicks, 24, su--ffered fatal injuries when the van he was asleep in w--as hit by Mr Scott's Audi A6. Maj Scott denies a ch--arge of causing death by careless driving. Prosecutor Charles Gabb alleged the defendant, from Green Lane in Shepperton, Surrey, had crossed the carriageway of the 60mph-limit B390 in Shrewton near Amesbury. T--he weather was "awful" and there was strong wind and rain, he told jurors. He said Mr Scott's car was descri--bed as "twitching" and "may have been aquaplaning" before striking the first vehicle; a BMW driven by Cra--ig Reed. Mr Scott's Audi then returned to his side of t--he road but crossed the carriageway again before colli--ding | No paraphraser | Maj Richard Scott, 40, is accused of driving at speeds of up to 95mph (153km/h) in bad weather before the smash on a B-road in Wiltshire. Gareth Hicks, 24, su--ffered fatal injuries when the van he was asleep in w--as hit by Mr Scott's Audi A6. Maj Scott denies a ch--arge of causing death by careless driving. Prosecutor Charles Gabb alleged the defendant, from Green Lane in Shepperton, Surrey, had crossed the carriageway of the 60mph-limit B390 in Shrewton near Amesbury. T--he weather was "awful" and there was strong wind and rain, he told jurors. He said Mr Scott's car was descri--bed as "twitching" and "may have been aquaplaning" before striking the first vehicle; a BMW driven by Cra--ig Reed. Mr Scott's Audi then returned to his side of t--he road but crossed the carriageway again before colli--ding head-on with a Ford Transit van in | 0.0031 |

# G   Effectiveness of Ensembling Detectors from RADAR

We study the ensembling detection performance by combining two RADAR-detectors' prediction probability and report the detection AUROC calculated using the combined prediction probability. The combined prediction probability $E(A, B, \beta, x) = (1 - \beta)\mathcal{D}_\phi^A(x) + \beta\mathcal{D}_\phi^B(x)$ is a weighted sum of the prediction probability of the base model $A$ and the augmented model $B$. The detection performances are shown in Table A6. We explore various ensembling ratios and ensembling models. Ensembling with the base model itself and setting the ensembling ratio to 0 both mean no ensembling. Setting the ensembling ratio to 1 is another extreme case, which refers to the transfer detection scheme mentioned in Section 4.3. From the results, we can see that the ensembling detection's effectiveness can be influenced by both the ensemble model and the ensemble ratio.

Table A6: Ensembling Detection results averaged on 4 datasets. Ensembling detection means combining base model $A$'s output prediction and augmented model $B$'s output prediction to detect $A$'s generation. The ensemble ratio $\beta$ varies from 0 to 1. Red text represents a better detection AUROC than that of no-ensembling

| Base Model | Ensemble Ratio | Augmented Model | | | |
|---|---|---|---|---|---|
| | | Vicuna-7B | Dolly-V1-6B | Camel-5B | Dolly-V2-3B |
| Vicuna-7B | 0 | 0.906 | | | |
| | 0.5 | 0.906 | 0.861 | 0.866 | 0.814 |
| | 1 | 0.906 | 0.822 | 0.749 | 0.684 |
| Dolly-V1-6B | 0 | 0.859 | | | |
| | 0.5 | 0.884 | 0.859 | 0.846 | 0.829 |
| | 1 | 0.877 | 0.859 | 0.722 | 0.764 |
| Camel-5B | 0 | 0.920 | | | |
| | 0.5 | 0.929 | 0.926 | 0.92 | 0.897 |
| | 1 | 0.882 | 0.883 | 0.92 | 0.764 |
| Dolly-V2-3B | 0 | 0.769 | | | |
| | 0.5 | 0.803 | 0.763 | 0.753 | 0.769 |
| | 1 | 0.794 | 0.727 | 0.642 | 0.769 |

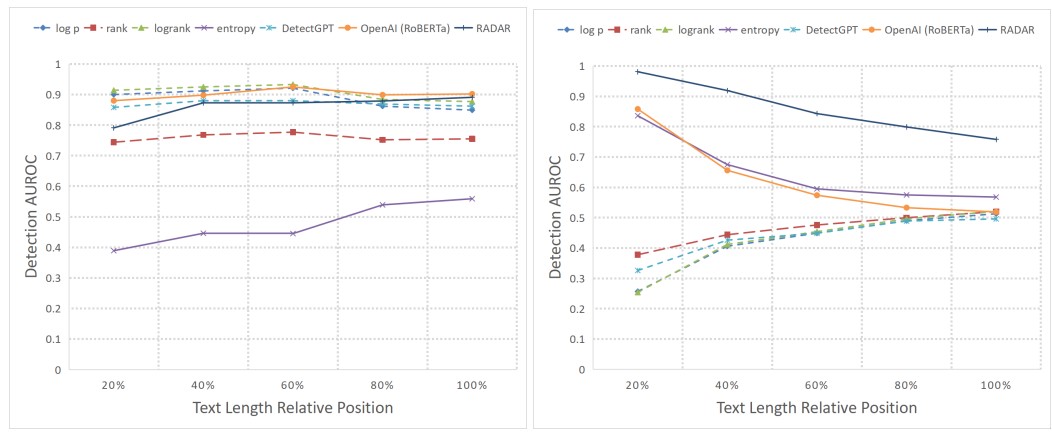

(a) w/o paraphraser      (b) RADAR-Unseen paraphraser

Figure A2: Detection AUROC of RADAR toward AI-generated texts with different lengths. The reported value is the AUROC of 8 LLMs averaged over 4 human-text datasets. The value on the X-axis means the texts' relative length rank position in the whole evaluation dataset. For example, 20% means the shortest 20% sentences.

## H   Text Detection Towards Different Lengths

We study RADAR's effects on AI-generated texts with different lengths. We grouped the evaluation dataset into 5 subsets according to the length of the AI-text. The results are shown in Figure A2. We summarize our observations below:

- For the group {log probability, rank, log rank, DetectGPT}, without paraphrasing, these methods are not really sensitive to the length of the text. When facing paraphrasing, however, their performance increases with a longer text length.

- For the group {entropy, OpenAI (RoBERTa), RADAR}, without paraphrasing, these methods have a better detection performance for longer texts. On the contrary, their performance degrades when facing longer paraphrased AI-text (even though RADAR seems much better for short-text detection, it still outperforms other methods by a large margin, see Figure A2b).

# I  Detection on Dipper Paraphrasing

We also explore the use of RADAR to detect other advanced paraphrasers. We use Dipper proposed in [18] (L60-O60 version) to paraphrase the evaluation dataset following the same setup of RADAR-Unseen paraphrasing and RADAR-Seen paraphrasing. The results are shown in Figure A3. We can see from the results that the green bars are much higher than the red bars except for DetectGPT, which means in general, Dipper is less destructive than GPT-3.5-Turbo to these detectors. RADAR's detection AUROC on Dipper reaches 0.9. One thing to be noted is that though the OpenAI (RoBERTa) can perform well under Dipper paraphrasing, it still cannot be regarded as robust because it can be bypassed by other paraphrasers (RADAR-Seen and RADAR-Unseen).

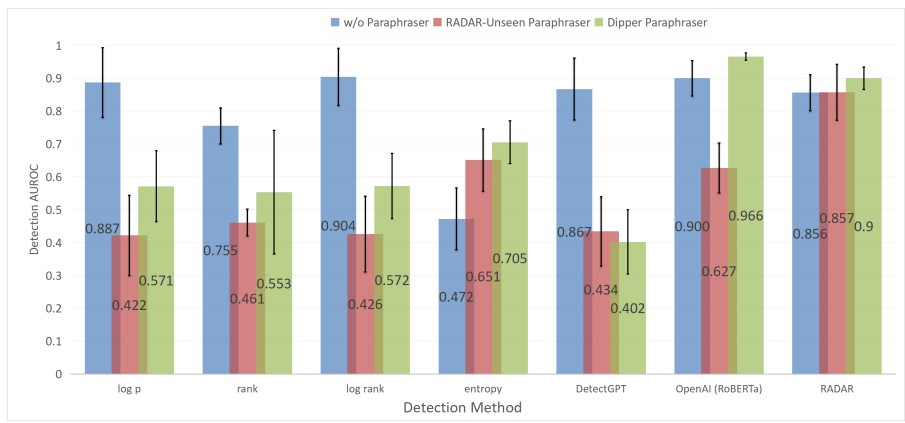

Figure A3: Augmented version of Figure 2. The added green bar represents RADAR's detection performance under Dipper paraphrasing.

# J  Sensitivity Analysis of RADAR on the Hyperparameter $\lambda$

Please refer to our discussion in Section 4.6.

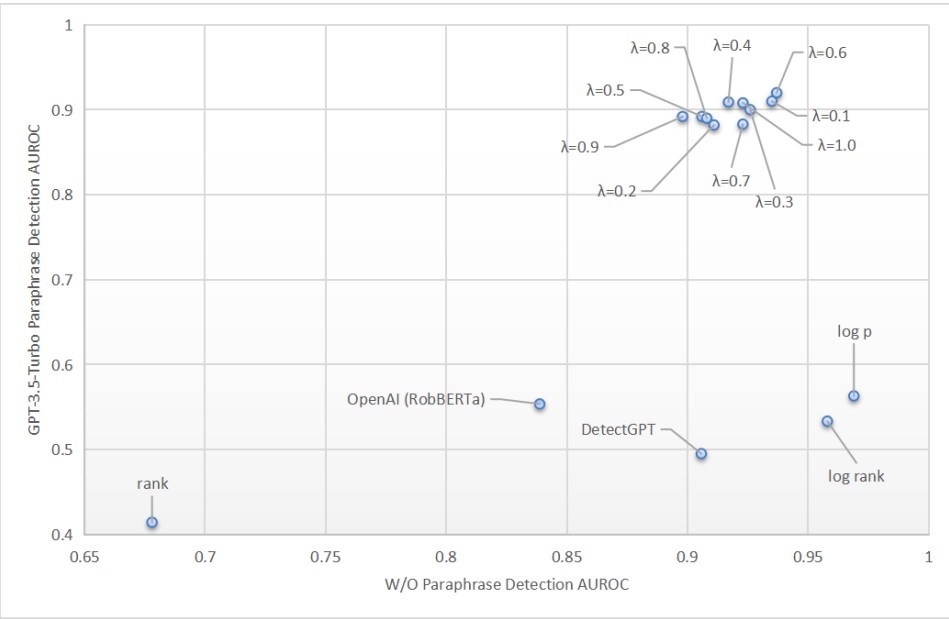

Figure A4: Sensitivity analysis for $\lambda$ in Eq. (3). The RADAR models are trained with Vicuna-7B with different $\lambda$ values ranging from 0.1 to 1.0. We then compare these RADAR detectors with other baselines on Vicuna-7B's 4 evaluation datasets. The horizontal/vertical axis shows the AUROC of unperturbed/paraphrased AI-text.

# K Detection on GPT-4 Generated Texts

Figure A5 shows RADAR's transfer detection for GPT-4. We use RADAR detectors trained on weaker LLMs (Vicuna-7B, Camel-5B, etc.) to detect the generation of GPT-4. We prompted GPT-4 with the instruction: *You are a helpful assistant to complete given text:*, to generate texts based on the same evaluation datasets (Xsum, SQuAD, WP and TOFEL) used when evaluating other LLMs. The detection performance have been discussed in Section 4.3.

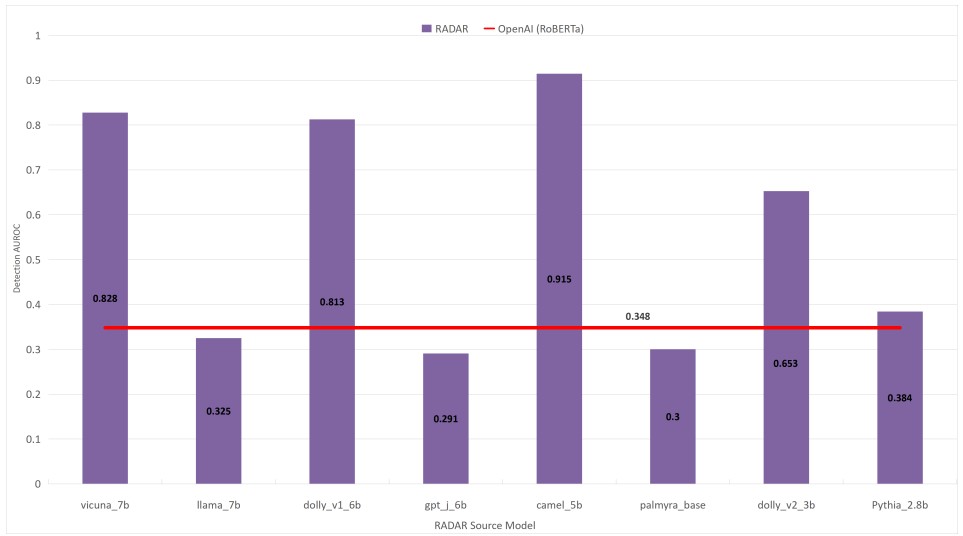

Figure A5: RADAR detection performance on GPT-4. The detector is trained with a source model, and the evaluation is measured with the texts generated from GPT-4.

# L Use GPT-3.5 to Assess RADAR-paraphrasers

After RADAR training, we not only get a detector but also a paraphraser. We use GPT-3.5-Turbo to assign a score to these paraphrasers' generation to assess the language generation capability of these paraphrasers and compare with their initial version (T5-large) to see how adversarial training benefits them. We compare these paraphrasers on a WebText subset with 100 samples. For each sentence in this subset, we first use GPT-2-XL to generate one sentence, and then use the 5 paraphrasers (T5-large and 4 paraphrasers trained on 4 instruction-tuned models' generation) to paraphrase this sentence respectively. Then we input these 5 paraphrased sentences combining an instruction to GPT-3.5-Turbo. Our instruction is *You are given an array of five sentences. Please rate these sentences and reply with an array of scores assigned to these sentences. Each score is on a scale from 1 to 10, the higher the score, the sentence is written more like a human. Your reply example: [2,2,2,2,2].* Then we sniff the score for each sentence from the returned answer and make the comparison. The results have been discussed in Section 4.5.

