# OpenReview forum: "RADAR: Robust AI-Text Detection via Adversarial Learning"
_NeurIPS.cc/2023/Conference — NeurIPS 2023 poster_

### Official Review · Reviewer_acyW · 2023-06-30

**Soundness:** 4 excellent
**Presentation:** 3 good
**Contribution:** 3 good
**Rating:** 6
**Confidence:** 4

**Summary:**

The paper proposes a framework called RADAR for training a robust AI-text detector using adversarial learning. The framework consists of a paraphraser and a detector, both based on large language models (LLMs). The paraphraser aims to generate realistic contents to evade AI-text detection, while the detector aims to enhance AI-text detectability. The two players iteratively update their model parameters until their respective validation loss becomes stable. Experimental results show that RADAR outperforms existing AI-text detection methods, especially when paraphrasing is involved. The paper also highlights the transferability of RADAR's detection capability to other LLMs.

**Strengths:**

+ The paper brings back the success of adversarial learning. The idea of employing an adversarial AI text paraphraser is simple but reasonable.
+ The experiments conducted on 8 different LLMs and 4 datasets demonstrate the effectiveness of RADAR in achieving consistently high detection performance.
+ The proposed method is not only effective with the seen paraphraser but also generalizable across different unseen paraphrser.

**Weaknesses:**

+ The generation quality after post-processed by the AI paraphraser is my major concern. Since the paraphrase model is inferior (T5 in your case) to the generation model, I suspect that the machine-generated text degrades in quality after paraphrasing. Meanwhile, we do not know whether the paraphrased text keeps its semantics unchanged. Therefore, I suggest the authors add an experiment comparing the text before and after being paraphrased.
+ The parameter scale of studied language models in the experiment is at most 7b.  I am curious to see the performance on larger models like GPT3.5 or GPT4. I understand that the cost of open AI API is one of the barriers to conducting such experiments. If this is the case, you may perform experiments on a sampled subset from the datasets to save the budget.
+ In line 268 you claim that all existing methods suffer from a notable performance drop on paraphrased AI text. However, from Figure 4(a) and Figure 2, we could observe that the entropy method actually becomes better.

**Questions:**

+ Have you ever tried the paraphraser trained by [1]?  I want to see whether your proposed RADAR could be spoofed by this paraphraser.



[1]Paraphrasing evades detectors of AI-generated text, but retrieval is an effective defense

**Limitations:**

The limitations are adequately discussed.

---

> ### Author Rebuttal · Authors · 2023-08-08
>
> We thank the reviewer for the time and expertise you have invested in these reviews. We are delighted to receive the positive feedback that our method is simple but effective.
>
> ---
> * **Comment 1**: The generation quality after post-processed by the AI paraphraser is my major concern. Since the paraphrase model is inferior (T5 in your case) to the generation model, I suspect that the machine-generated text degrades in quality after paraphrasing. Meanwhile, we do not know whether the paraphrased text keeps its semantics unchanged.
>
>   **Response**: Thank you for pointing out this issue. In our evaluation, we not only use T5 as the paraphraser (RADAR-Seen) but also use GPT-3.5-Turbo as the paraphraser (RADAR-Unseen). The experimental results showed that no matter whether the paraphraser is inferior (T5) to the generation model or superior (GPT-3.5-Turbo) to the generation model, RADAR can achieve good performance.
> 	In terms of the paraphrased texts’ quality, following the suggestion from reviewer M6sT, we showed the RADAR-Seen paraphraser’s iBLEU score is much higher than the original T5-large on the standard benchmark: Quora Question Pairs. The iBLEU score is shown in **Fig. R2** in the attached PDF. We also note that in RADAR training, we take the human-text, the original AI-text (by a target LLM), and the paraphrased AI-text (by a paraphraser) into consideration. Therefore, the detector not only sees the paraphrased texts.
>
> ---
> * **Comment 2**: I am curious to see the performance on larger models like GPT3.5 or GPT4. I understand that the cost of open AI API is one of the barriers to conducting such experiments. If this is the case, you may perform experiments on a sampled subset from the datasets to save the budget.
>
>   **Response**: Thanks for your understanding that training a detector for those OpenAI close-sourced LLMs is expensive for us. Due to the limit (Quota & Tokens-Per-Minute & Requests-Per-Minute) of accessing OpenAI API service, training a detector for GPT-3.5/GPT-4 is both time-consuming and money-consuming, and honestly beyond our capability (for example, we estimated that the complete RADAR training process for GPT-4 will cost at least $1000). The training data generation process will also be dramatically slow due to the TPM and RPM limit. However, evaluation is feasible. We use RADAR detectors that are trained for other language models to perform transfer detection on GPT-4’s generation. The detection results are shown in **Fig. R4** in the attached pdf. The results show that 5 RADAR models can outperform the Roberta-based detector, and three of them can achieve 0.8+ detection AUROC. The results show that RADAR can achieve good detection for GPT-4.
>
> ---
> * **Comment 3**:  However, from Figure 4(a) and Figure 2, we could observe that the entropy method actually becomes better.
>
>   **Response**: This observation is correct. We also observed this phenomenon and in Sec. 4.2 we have mentioned that although entropy is robust to paraphrasing, its detection AUROC is quite low no matter whether the AI-texts are paraphrased or not.
>
>   Back to line 268, where we write "On paraphrased 268 AI-text, all existing methods suffer from a notable performance drop." Actually, according to the context, this sentence is correct. Because this sentence is to depict the recursive paraphrase attack to AI-text and from the corresponding figure (Fig. 4. a), we can see that after the 2nd paraphrase, the detection auroc of entropy begins to degrade indeed. We will make this discussion clearer to avoid confusion.
>
>   In summary, even if the entropy method becomes better after 1 round paraphrase, it is still not a good detector because of the low AUROC and it is not robust to recursive attacks.
>
>
> ---
> * **Comment 4**: Have you ever tried the paraphraser trained by [1]? I want to see whether your proposed RADAR could be spoofed by this paraphraser.
>
>   **Response**: We thank the reviewer for pointing us to a new paraphraser. Following your suggestion, we use Dipper L60-O60 version to perform a paraphrase attack to the evaluation dataset following the same setup of RADAR-Unseen attack and RADAR-Seen attack. We listed RADAR and other baselines’ performance below.
>   |Detection Method|log p|rank|log rank|entropy|DetectGPT|Roberta-based Detector|RADAR|
>   |:-----|:----:|:----:|:----:|:----:|:----:|:----:|:----:|
>   |**w/o Paraphraser Detection AUROC**|0.887|0.755|0.904|0.472|0.867|0.900|0.856|
>   |**GPT-3.5-Turbo(RADAR-Unseen) Paraphraser Detection AUROC**|0.422|0.461|0.426|0.651|0.434|0.627|0.857|
>   |**Dipper Paraphraser Detection AUROC**|0.571|0.553|0.572|0.705|0.402|0.966|0.901|
>   |**RADAR-Seen Paraphraser Detection AUROC**|0.198|0.284|0.209|0.820|0.159|0.806|0.922|
>
>   We can see from the table that the values in the third line are much higher than the second line except for DetectGPT, which means Dipper is less destructive than GPT-3.5-Turbo to the detection system listed in the table.  RADAR’s performance under this attack reaches 0.9. One thing to be noted is that though Roberta-based detector can perform well under Dipper Paraphrasing, it still can not be regarded as robust because it can be bypassed by other paraphrasers.

---

> > ### Comment · Area_Chair_tw6v · 2023-08-19
> > **Thanks for your response**
> >
> > Dear authors,
> >
> > Thanks for your detailed response. I think your response has addressed some of the reviewer's concerns.
> >
> > Best,
> >
> > AC

---

> ### Comment · Reviewer_acyW · 2023-08-19
> **Thanks for your response**
>
> My concerns are addressed after reading the rebuttal response from the authors. So I adjust my recommendations accordingly.

---

> > ### Author Response · Authors · 2023-08-19
> >
> > We thank the reviewer for the positive feedback and for the constructive comments.

---

### Official Review · Reviewer_uRZ4 · 2023-07-04

**Soundness:** 3 good
**Presentation:** 3 good
**Contribution:** 3 good
**Rating:** 5
**Confidence:** 3

**Summary:**

The paper introduces a new framework called RADAR, which aims to address the gap in existing AI-text detection methods. RADAR utilizes adversarial learning to jointly train a robust AI-text detector. The framework consists of a paraphraser and a detector, both trained through adversarial training. The paraphraser's objective is to generate realistic content that can evade AI-text detection. RADAR continuously updates the paraphraser and the detector based on the feedback received from each other. The framework is evaluated using eight different language models (Pythia, Dolly 2.0, Palmyra, Camel, GPT-J, Dolly 1.0, LLaMA, and Vicuna) across four datasets. The experimental results demonstrate that RADAR outperforms existing AI-text detection methods, particularly when paraphrasing techniques are employed.

**Strengths:**

1. The detection of ChatGPT generated texts is an interesting and important research problem.

2. The proposed method is reasonable.

3. Experimental results seem promising.

**Weaknesses:**

1. It is not clear whether the proposed method can achieve good result if the LLM for text generation (usually black-box) is much power than the paraphraser used in the proposed method (open-sourced).

2. The proposed method is not tested on the texts generated by powerful LLMs like GPT-4.

3. The experimental settings are a little confusing.

**Questions:**

1. The experimental settings are a little confusing. What are the training data? Why is the test data constructed in such a naive way as "for the w/o paraphrasing setting, we use the original AI-text corpus M generated by a target LLM". Why not construct some domain and task-specific AI texts?

2. Why not do experiments on SOTA LLMs like GPT-4 and ChatGPT?

**Limitations:**

The authors have addressed the limitations.

---

> ### Author Rebuttal · Authors · 2023-08-08
>
> We thank the reviewer for the time and expertise you have invested in these reviews. We are delighted to receive the positive feedback that our research problem is both interesting and important, the proposed method is reasonable, and the experimental results are promising.
>
> Below we provide point-by-point responses to your comments and questions.
> * **Comment 1**: It is not clear whether the proposed method can achieve good result if the LLM for text generation (usually black-box) is much power than the paraphraser used in the proposed method (open-sourced).
>
>   **Response**: Thanks for the reviewer's suggestion. We believe that we have discussed this setting in our paper.
>   Below we show a detailed discussion in different scenarios to address the reviewer’s concern.
>   1. When LLM is much more powerful than the paraphraser used in the RADAR training: Though we use a small-size paraphraser (T5-large, 770M parameters) to train a RADAR detector, our detector showed effectiveness against many powerful LLMs (from 3B to 7B) in Fig. 3. Though the LLMs in Fig.3 are open-sourced, they can be regarded as black-box as the LLM to be detected in the transfer setting (i.e. the off-diagonal entries in the matrix) is oblivious in the training process of the detector.
>   2. When LLM is much more powerful than the paraphraser used in the RADAR evaluation. We need to emphasize that we not only introduced RADAR-Unseen paraphraser (GPT-3.5-Turbo, 175B parameters) but also introduced RADAR-Seen paraphraser (T5-large, 770M parameters). RADAR showed great detection performance in both cases. (See Table 2 for more details).
>
>   In addition, we also test RADAR on the SOTA LLM: GPT-4 (black box) following your suggestion. As training a RADAR detector for GPT-4 is very expensive, we perform transfer detection on GPT-4 using RADAR detector trained for smaller LLMs, e.g. Vicuna-7B, Camel-5B, etc. We reported the results and the analysis in the response to the reviewer’s 2nd comment.
> ---
> * **Comment 2**: The proposed method is not tested on the texts generated by powerful LLMs like GPT-4. & Why not do experiments on SOTA LLMs like GPT-4 and ChatGPT?
>
>   **Response**: Thank you for the suggestion. In fact, we indeed tried to incorporate ChatGPT/GPT-4 into RADAR’s target list before the submission. But there are many reasons that we don’t either train a detector targeting them or just evaluate our method’s performance on their generation.
>
>   Firstly, at that time, we have limits (Tokens-Per-Minute (TPM) and Requests-Per-Minute (RPM)) for ChatGPT’s API usage. The generation process will be dramatically slow due to the TPM and RPM limits. Moreover, we didn't get access to GPT-4 (On July 7th we have access to GPT-4’s API).
>
>   Secondly, even up to now, the user still can’t get useful decoding information of GPT-4/ChatGPT from the return content of the API call as mentioned in [R9]. The other baselines except for Roberta-based Detector can’t work if there isn’t some decoding information (for example, log probability).
>
>   Following the reviewer's suggestion, we performed transfer detection on GPT-4 which has been mentioned in the response to the 1st comment. The evaluation procedure is as same as Fig. 3. The detection results are shown in **Fig. R4** in the attached pdf. The results show that  5 RADAR models can outperform the Roberta-based detector, and three of them can achieve 0.8+ detection AUROC. The results show that RADAR can achieve good detection for GPT-4.
>
> ---
> * **Comment 3**: What are the training data?
>
>   **Response**: As briefly mentioned in Line 188, we have Human-text sampled from WebText [R10] and original AI-text generated by the corresponding LLM. During training, the AI-text will be perturbed. The Human-text, AI-text, and the perturbed AI-text will all be used to train the detector. We introduced the training data in Section 4.1 and in Appendix (Section A).
> ---
> * **Comment 4**: Why not construct some domain and task-specific AI texts?
>
>   **Response**: Thank you for the comment. In fact, the AI-texts in our evaluation dataset are already domain and task-specific. We choose  Xsum (news ), SQuAD (Academic Essays), Reddit WritingPrompts (Creative Writing Stories), and Tofel Essay (Non-native speaker's essay) as the human-texts in the evaluation dataset. We let the target LLM do text completion for these domain-specific or task-specific human-texts to get AI-texts, so the AI-texts can be regarded as domain and task-specific, too.
>
> ## References
> Please see **Rebuttal References** in the global response.

---

> > ### Comment · Reviewer_uRZ4 · 2023-08-18
> > **Thanks for the response**
> >
> > Thank the authors for their detailed response. I am satisfied with their response.

---

> > > ### Author Response · Authors · 2023-08-19
> > >
> > > We are delighted to learn that the reviewer finds our response satisfactory! Again, we thank the reviewer for your valuable feedback and comments.

---

### Official Review · Reviewer_M6sT · 2023-07-06

**Soundness:** 3 good
**Presentation:** 2 fair
**Contribution:** 3 good
**Rating:** 6
**Confidence:** 4

**Summary:**

This paper proposes an adversarial learning framework, called RADAR,
for improving the robustness of AI-text detector.  Given a target
large language model (LLM) that generates AI-text, the framework
trains a detector model and a paraphraser model both are also based on
an LLM simultaneously.  The performance of RADAR on AI-text detection
is evaluated through eight LLMs and four datasets.  The results show
that RADAR can achieve the best classification accuracy measured by
AUROC, when paraphraser is introduced to illude classification.  The
authors further clarified the relatively high versatility of detectors
trained through RADAR, especially when using LLMs fine-tuned through
instruction tuning.


**Strengths:**

- The paper presents a straightforward but effective way of training a
  robust AI-text detector, which is backed up by the empirical
  experiments.

- The authors also demonstrate that the detector trained by the
  proposed method can perform well also for AI-texts generated by
  another LLM, measuring the "transferability" across eight publicly
  available LLM, revealing an advantage of instruction tuning for LLM.


**Weaknesses:**

- One critical point is in line 11 of Algorithm 1: $x_{h}$ and $x_{m}$
  are sampled from $\mathcal{H}$ and $\mathcal{M}$ respectively, which
  means that there are arbitrary pairs of $(x_{h}, x_{m})$ drawn from
  $\mathcal{H}\times\mathcal{M}$.  I suspect this could be wrong and
  $x_{h}$ and $x_{m}$ must be corresponding, since $x_{p}$ is
  generated from $x_{m}$ in l.12 and the tuple of $x_{h}$, $x_{m}$,
  and $x_{p}$ is saved together in the batch in l.15, enabling an
  indirect contrastive learning.

- ll.165-166: What is the purpose of the normalization function $A$
  and how is it actually implemented?  $R(x_{p},\phi)$ is already
  between 0 and 1 according to Equation 1.

- It is nice to evaluate paraphrasers in Section 4.5, but what is the
  justification to choose this method?  Paraphrase generation methods
  are in general evaluated on paraphrase generation tasks, such as
  Quora.  Human evaluation is a possible alternative.  On the other
  hand, the process of using GPT-3.5 Turbo for rating is not clearly
  explained, and anyways it will not draw reliable results and
  consequent conclusions.

**Questions:**

- The numbers in Figures 2, 3, 4, and 5 are too small to read on US
  letter papers.

- Bars in Figure 2 should also have error bars, because the bars
  except for RADAR are the average of the AUROC for the four test
  sets.  The results for RADAR are also averaging eight LLMs (or 64
  combinations of training/testing LLMs) and four task results.
  The authors have some space to also show the worst and best models.

- The rows should be grouped by "Evaluation Schema" and the first two
  columns should be swapped.  This makes direct comparisons of
  different methods easier, also avoiding the colors for the schema,
  which are difficult to distinguish by people with achromatic vision.

- Coloring scheme for the two heat-maps in Figure 3 should be shared,
  in order to avoid misunderstanding based on the colors only.  Text
  color for dark cell should also be reconsidered.

- The paper has some language problems, such as present tense in
  Section 4, "pair" -> "paired" in l.101, and "did not participate" ->
  "has not participated" in l.211.


**Limitations:**

Although the authors indicated Section 4.2, the limitations of this
work is rather discussed in Section 4.3 through a discussion on
transferability.  The high values for transferability suggest the
proposed method and resulting AI-text detectors are versatile.
However, it does not necessarily stop a kind of arms race between LLM
and detector, as evidenced in Figure 3: detectors trained on weaker
LLMs perform much worse for test data generated by stronger LLM.

There is no discussion on the potential societal impact, but according
to the above note on the arms race, the results of AI-text detector at
some point would mislead users, potentially causing severe issues.

---

> ### Author Rebuttal · Authors · 2023-08-08
>
> We thank the reviewer for the time and expertise you have invested in these reviews. We are grateful to receive your appraisal that our work provides a straightforward but effective way of training a robust AI-text detector. This is indeed our original aspiration.
>
> We reply to individual points below:
> * **Comment 1**: I suspect this could be wrong and $x_h$ and $x_m$ must be corresponding.
>
>   **Response**: The reviewer’s understanding is correct: $x_h$, $x_m$ and $x_p$ are corresponding. We will update our presentation in the revised version of our paper to avoid misunderstanding.
> ---
> * **Comment 2**: What is the purpose of the normalization function $A$ and how is it actually implemented?
>
>   **Response**: Reward normalization is a typical technique adopted by reinforcement learning implementations [R1, R2], which will help to stabilize the training process. During training, we use Normalization to shift the distribution of the rewards. Mathematically, it can be expressed as below:
>
>   Given a buffer $\mathcal{B}$ with many samples ($n$ of them) and each holds a reward obtained from the detector: $R_1, R_2, \ldots, R_n$:
>   1. Calculate the mean ($\mu$):
>      $$\mu = \frac{\sum_{i=1}^{n} R_i}{n}$$
>   2. Calculate the standard deviation ($\sigma$):
>      $$\sigma = \sqrt{\frac{\sum_{i=1}^{n} (R_i - \mu)^2}{n}}$$
>   3. Get Advantages:
>      $$A_i = \frac{R_i - \mu}{\sigma}$$
>   Where $A_i$ represents the advantages used to calculate the policy gradient.
> ---
> * **Comment 3**: Paraphrase generation methods are in general evaluated on paraphrase generation tasks, such as Quora.
>
>   **Response**: This is a great suggestion to further validate the paraphrasers obtained from RADAR.  Following [R3, R4], we evaluated RADAR-paraphrasers on Quora Question Pairs [R5] and use iBLEU (α=0.8) [R6] as the metric (higher is better). The result is shown in **Fig. R2** in the attached pdf. We can conclude that based on the iBLEU metric, the paraphrasers can be improved via RADAR as all the RADAR-paraphrasers can achieve a larger iBLEU score than T5-large (which is the initialization of the paraphraser). We will add this result in the revised version to support our argument.
> ---
> * **Comment 4**: On the other hand, the process of using GPT-3.5 Turbo for rating is not clearly explained, and anyways it will not draw reliable results and consequent conclusions.
>
>   **Response**: We understand the reviewer’s concern. We hope that our additional evaluation on Quora (see our response for Comment 3) provides another evidence for our claim.  Our evaluation is motivated by the study from the Vicuna Team that used GPT-4 to compare Vicuna with other open-source LLMs back in March 2023 [R7] (we did not have access to GPT-4 at that point). We use GPT-3.5-Turbo to rate all the five paraphrasers’ generations together at the same time. We introduced in Appendix (Section I) the details about the evaluation instruction.
> ---
> * **Comment 5**: Comment on presentation ( font size, text color, coloring scheme, and table organization)
>
>   **Response**: We will incorporate these suggestions into our updated version.
> ---
> * **Comment 6**: Bars in Figure 2 should also have error bars because the bars except for RADAR are the average of the AUROC for the four test sets. The results for RADAR are also averaging eight LLMs (or 64 combinations of training/testing LLMs) and four task results. The authors have some space to also show the worst and best models.
>
>   **Response**: It appears that there is a misunderstanding. In fact, in Fig. 2 the reported values for all the detection methods are all averaged across 8 target LLMs. In total, our evaluation includes the AI-texts generated from 8 (LLMs) * 4 (datasets) combinations). More specifically, we follow the evaluation setup in [R8] to evaluate a target LLM. Under this setting, all the methods except for Roberta-based Detector (supervised trained) use the target LLM itself for detection. For example, RADAR uses the corresponding detector trained with the target LLM, and baselines such as DetectGPT directly use the target LLM to access decoding information.
>
>   We agree that Fig.2 should have error bars to reveal the difference between target LLMs. We draw error bars for all the detection methods using the detection standard deviation over 8 target LLMs and show the updated version in **Fig. R3** in the attached PDF. We also listed RADARs’ detection AUROC for different LLMs below:
>
>     |Target LLM|Pythia-2.8B|Dolly-V2-3B|Palmyra-Base|Camel-5B|GPT-J-6B|Dolly-V1-6B|LLaMA-7B|Vicuna-7B|
>     |:-----|:---:|:---:|:---:|:---:|:---:|:---:|:---:|:---:|
>     |**w/o Paraphrase Detection AUROC**|0.854|**0.769(Worst)**|0.916|**0.920(Best)**|0.820|0.859|0.808|0.906|
>     |**RADAR-Unseen Paraphrase  Detection AUROC**|0.866|0.876|0.861|**0.973(Best)**|0.757|0.923|**0.712(Worst)**|0.892|
> ---
> * **Comment 7**: However, it does not necessarily stop a kind of arms race between LLM and detector, as evidenced in Figure 3: detectors trained on weaker LLMs perform much worse for test data generated by stronger LLM.
>
>   **Response**: The reviewer’s observation is correct. We totally agree that AI-text detection can be an arms race, and therefore adversarial learning is a natural approach to tackle this challenge. We would like to reemphasize that our RADAR is a general framework that applies to any LLM. Whenever a new and more powerful LLM emerges, we can use RADAR to obtain a better detector.
>
> ---
> * **Comment 8**: The results of AI-text detector at some point would mislead users, potentially causing severe issues.
>
>   **Response**: Although our detector is not perfect (and so are existing detectors), it shows better robustness than existing methods against paraphrasing, and we believe RADAR can be a useful tool to assist users to identify AI-written content at scale and help mitigate issues related to AI-written content.
>
> ## References
>   Please see **Rebuttal References** in the global response.

---

> > ### Comment · Reviewer_M6sT · 2023-08-15
> > **Read the response**
> >
> > Thanks for confirming my comments in your response and spotting my
> > misunderstanding regarding Figure 2.  Most parts of your reply answer
> > my comments and questions.
> >
> > However, using GPT-3.5 Turbo for "evaluating your method" has not yet
> > been justified.  The authors cite a blog post but the contents have
> > never reviewed and justified in a scientific manner.  In my opinion,
> > automatic evaluation metric must be transparent and reproducible.
> > However, the method described in Appendix I does not guarantee them
> > and thus it does not qualify for evaluation.  As such, the results on
> > Quora are only the evidence that the proposed paraphrasers work well
> > and they should replace the results based on GPT-3.5 Turbo.

---

> > > ### Author Response · Authors · 2023-08-15
> > > **Thank you for your response**
> > >
> > > We thank the reviewer for the prompt response and for the suggestion of comparing paraphrasing quality on Quora, which is a brilliant idea! In the future version, we will replace Fig. 5 with the analysis on Quora, as the reviewer suggested.

---

> > > > ### Comment · Reviewer_M6sT · 2023-08-19
> > > > **Thank you for your understanding**
> > > >
> > > > If one wants to claim the performance of paraphraser alone, it is straightforward to evaluate it on a paraphrase generation task, and thus it is not that brilliant. Quora is merely one example of such tasks containing only rather short questions; it could not be appropriate considering the intended purposes of the paraphraser the authors have built and there could be datasets that better fit it.  Just for your information.

---

> > > > > ### Author Response · Authors · 2023-08-19
> > > > >
> > > > > We thank the reviewer's feedback! Indeed, our main proposal is the robust AI-text detector. The RADAR's paraphraser is a byproduct of our proposed adversarial training framework. Our analysis of the RADAR's paraphraser was meant to show that the paraphraser had some differences from its original model. We understand that to claim the paraphraser is better (which is not our intention) requires more rigour studies and evaluations. We will make this point clearer in the revised version.

---

### Official Review · Reviewer_8fAM · 2023-07-06

**Soundness:** 3 good
**Presentation:** 3 good
**Contribution:** 3 good
**Rating:** 6
**Confidence:** 3

**Summary:**

This paper proposes RADAR, a detection method for AI-generated text that is based on adversarial learning. Inspired by the conceptual idea of generative adversarial networks, RADAR consists of a paraphraser and a detector module. The former aims to generate textual content that evades an AI content detection system, while the latter aims to detect AI-generated text. Both modules are based on LLMs and trained end-to-end. Once trained, only the detector is then used for evaluation.
The authors conduct experiments with 8 LLMs (incl. Pythia-2.8B, GPT-J-6B, LLaMA-7B) across 4 datasets (Xsum, SQuAD, Reddit WritingPrompts, TOEFL-dataset). RADAR is compared against a range of detection baselines in multiple settings (with and without paraphrasing of test set texts). Experimental results demonstrate that RADAR attains a slightly lower performance on tasks without paraphrasing as compared to baselines, but outperforms all existing works when the texts are paraphrased. The authors provide more extensive results on evaluating RADAR detection performance across all possible pairings of the 8 LLMs when one of the models is used for paraphrasing during training and the other one during testing. Interestingly, the paper also shows that RADAR is robust against paraphrasing a sequence multiple times (in contrast to the baselines).

**Strengths:**

* The paper is very well-written, and the technical aspects are well-explained. The analysis of RADAR is thorough, and the experimental results are mostly promising.
* The proposed algorithm is novel and the idea of adversarial learning in the context of AI-generated text is interesting.
* Overall, I think that this method and work represent a solid contribution to the field.


**Weaknesses:**

* In terms of model performance, RADAR shows to be outperformed by existing baselines on original (i.e., unperturbed) data to a notable extent. While this is not necessarily problematic, it might be good to show some additional analyses further investigating how this tradeoff can be mitigated. For example, how does adjusting the weight coefficients in Equation 3 impact the detection results? Is there a way to better optimize these parameters, to ensure that model accuracy on unperturbed data remains competitive with baselines?
* Missing discussion and limitations: The paper does not list any possible limitations, nor does it have a discussion section to contextualize the findings better into the existing literature. Since detecting AI-generated text has a direct application in more practical settings, it would have been helpful to address how well RADAR can be employed by users and practitioners, and which challenges remain. Moreover, the authors do not discuss potential ethical considerations arising from their work.



**Questions:**

* Can you elaborate on the dataset sizes and splits that you used to evaluate RADAR? This is a technical detail, but it would be helpful to get an estimate of test set sizes to better interpret the results in Table 2.
* Did you run any additional experiments analyzing the model tradeoff between performance on paraphrased vs non-paraphrased data?


**Limitations:**

The authors neither discuss the limitations of their work nor do they mention ethical considerations arising from these results. I strongly encourage the authors to address these two missing points.

---

> ### Author Rebuttal · Authors · 2023-08-08
>
> We thank the reviewer for the time and expertise you have invested in these reviews. We are delighted to receive the positive feedback that our work provides a solid contribution to the field, especially that the paper is well-written, the technical aspects are well-explained, the analysis is thorough, the experiments are mostly promising, and the proposed algorithm is novel.
>
> Below we provide point-by-point responses to your comments and questions.
>
> * **Comment 1**: Did you run any additional experiments analyzing the model tradeoff between performance on paraphrased vs non-paraphrased data?
>
>   **Response**: This is a great question. Following your advice, to study the model trade-off, we conduct a sensitivity analysis on the coefficient $\lambda$ in Eq.3 (we select 10 different $\lambda$ values in the range of 0 to 1, and the original reported $\lambda$ value is 0.5). The result is shown in **Fig. R1** in the attached PDF, the horizontal/vertical axis in **Fig. R1** shows the AUROC of unperturbed/paraphrased AI-text of each detector (including RADAR and baselines). We also list the exact values (AUROC) below.
>
>   | $\lambda$ | 0.1 | 0.2 |0.3 | 0.4 |0.5 | 0.6 |0.7 | 0.8 |0.9 | 1.0 |
>   | :-----| :----: | :----: |:----: | :----: |:----: | :----: |:----: | :----: |:----: | :----: |
>   | **w/o Paraphrase Detection AUROC**| 0.935 | 0.911 | 0.926 | 0.917 | 0.906 | 0.937 | 0.923 | 0.908 | 0.908 | 0.923 |
>   | **RADAR-Unseen (GPT-3.5-Turbo) Paraphrase Detection AUROC** | 0.910 | 0.882 | 0.900 | 0.909 | 0.892 | 0.920 | 0.883 | 0.890 | 0.892 | 0.908 |
>
>   Due to the constraint in rebuttal time and training cost, we only reported the model tradeoff of RADAR detector trained with Vicuna-7B.  We evaluated these RADAR detectors along with other baselines on the generated data of Vicuna-7B. We can see the stability of RADAR as the RADAR detectors are clustered in a region where both the two AUROC values are high. By tuning $\lambda$, we can promote RADAR’s performance on unperturbed data while still preserving high detection auroc on paraphrased data (for example, $\lambda$=0.6). Though there is a small gap in the w/o paraphrase detection performance of RADAR and the best baseline, RADAR shows significantly improved detection performance against paraphrased AI-text. The reviewer’s intuition is also correct that by carefully tuning $\lambda$, the tradeoff can be further mitigated. We will include this model tradeoff analysis of RADAR in the revised version.
> ---
> * **Comment 2**: Can you elaborate on the dataset sizes and splits (of test set sizes) that you used to evaluate RADAR?
>
>   **Response**: We briefly introduced the evaluation dataset in Section 4.1 and provided detailed information (including data sizes and splits) in Appendix (Section A). In short, we selected 500 samples from Xsum-train, 312 samples from SQuAD-train, 500 samples from WritingPrompts-train, and 90 samples from TOFEL-test as human-text. Then we employed 8 LLMs to generate the same number of AI-text counterparts as described in Line 138. Our evaluation dataset consists of both the AI-text and the human-text.
> ---
> * **Comment 3**: The authors neither discuss the limitations of their work nor do they mention ethical considerations arising from these results.
>
>    **Response**: Thank you for the comment. As other reviewers have pointed out, we addressed the limitation of RADAR in the observed slight reduction of detection performance when compared to the best detector in the without paraphrasing case, in order to obtain a significant gain in detecting AI-text with paraphrasing (line 225-228). Moreover, like every existing AI-text detector, our detector is not perfect and will likely give incorrect predictions in some cases. In the revised version, we will make a new paragraph of "Limitation" at the end of the Conclusion section to make these discussions more explicit. For potential ethical considerations, we will add that our method involves the use of existing large language models and may inherit the shortcomings therein. Also, users can use our tool to assist with identifying AI-written content at scale, but further validation steps are needed if the result is used as evidence.

---

> > ### Comment · Reviewer_8fAM · 2023-08-15
> > **Rebuttal acknowledged**
> >
> > Thanks to the authors for providing such a detailed response (esp. with the additional results re. Comment 1). I recommend that the authors add these additional results and limitations to the paper/appendix and, as already indicated with my rating, am in favor of accepting this work.

---

> > > ### Author Response · Authors · 2023-08-15
> > > **Thank you for your response**
> > >
> > > We thank the reviewer for your prompt response and for the valuable comments. We will certainly include all the new results and discussions presented during the author-reviewer discussion phase in the future version of our paper.

---

### Author Rebuttal · Authors · 2023-08-08

## Rebuttal References

[R1] PyTorch actor-critic example. (we are not allowed to post any link, so please google it to find the link)

[R2] PyTorch REINFORCE example. (we are not allowed to post any link, so please google it to find the link)

[R3] Tom Hosking and Mirella Lapata. Factorising Meaning and Form for Intent-Preserving Paraphrasing. In ACL/IJCNLP (Volume 1: Long Papers), pages 1405--1418, 2021.

[R4] Tom Hosking, Hao Tang and Mirella Lapata. Hierarchical Sketch Induction for Paraphrase Generation.  In ACL (Volume 1: Long Papers), pages 2489--2501, 2022.

[R5] Kaggle competitions on Quora Question Pairs. (we are not allowed to post any link, so please google it to find the link)

[R6] Hong Sun and Ming Zhou. Joint learning of a dual SMT system for paraphrase generation.  In ACL (Volume 2: Short Papers), pages 38--42, 2012.

[R7] The Vicuna Team. Vicuna: An Open-Source Chatbot Impressing GPT-4 with 90%* ChatGPT Quality. In LMSYS Org's blog, 2023. (we are not allowed to post any link, so please google it to find the link)

[R8] Eric Mitchell, Yoonho Lee, Alexander Khazatsky, Christopher D. Manning, and Chelsea Finn. Detectgpt: Zero-shot machine-generated text detection using probability curvature. CoRR, abs/2301.11305, 2023.

[R9] OpenAI. GPT-4 API general availability and deprecation of older models in the Completions API.  In OpenAI's blog, 2023. (we are not allowed to post any link, so please google it to find the link)

[R10] The project page of Open WebText Corpus. (we are not allowed to post any link, so please google it to find the link)

## Rebuttal Figures

Please download the attached PDF file.

---

### Comment · Area_Chair_tw6v · 2023-08-13
**Please have a look at the authors' response**

Dear reviewers,

Thanks for your constructive reviews. Looks like most of your reviews are really positive. The authors have submitted their responses to your reviews and answered your questions. Could you do us a favor and check if they've addressed your concerns?

Appreciate your help!

AC

---

### Decision · Program_Chairs · 2023-09-21

**Decision:**

Accept (poster)

**Comment:**

The paper centers around AI-Text Detection, a prominent topic within the realm of LLM. Reviewers find the proposed algorithm to be novel, and the idea of applying adversarial learning in the context of AI-generated text is interesting. The experiments conducted on various LLMs and datasets demonstrate that the proposed approach consistently achieves high detection performance. However, reviewers have raised concerns regarding the limitations statement, generation quality after post-processing by the AI paraphraser, the experimental setting, and the analysis of how the parameters impact the detection results. The author's responses have partially addressed the concerns raised by the reviewers, leading some to subsequently adjust their scores. Consequently, I am inclined to recommend acceptance.